# OPEN

# Identifying patterns in amyotrophic lateral sclerosis progression from sparse longitudinal data

Divya Ramamoorthy [1], Kristen Severson[2], Soumya Ghosh [2], Karen Sachs[1,3], Answer ALS*, Jonathan D. Glass [4], Christina N. Fournier[4], Pooled Resource Open-Access ALS Clinical Trials Consortium*, ALS/MND Natural History Consortium, Todd M. Herrington[5,6], James D. Berry [5], Kenney Ng [2] and Ernest Fraenkel [1]✉

The clinical presentation of amyotrophic lateral sclerosis (ALS), a fatal neurodegenerative disease, varies widely across patients, making it challenging to determine if potential therapeutics slow progression. We sought to determine whether there were common patterns of disease progression that could aid in the design and analysis of clinical trials. We developed an approach based on a mixture of Gaussian processes to identify clusters of patients sharing similar disease progression patterns, modeling their average trajectories and the variability in each cluster. We show that ALS progression is frequently nonlinear, with periods of stable disease preceded or followed by rapid decline. We also show that our approach can be extended to Alzheimer's and Parkinson's diseases. Our results advance the characterization of disease progression of ALS and provide a flexible modeling approach that can be applied to other progressive diseases.

Amyotrophic lateral sclerosis (ALS) is a neurodegenerative disease with a complex pathophysiology resulting in heterogeneous symptoms and progression[1,2]. The median length of survival from symptom onset is approximately three years; however, some individuals survive decades with the disease[3]. Longitudinal functional clinical metrics have gained widespread use as a tool to measure ALS progression. These clinical metrics, such as the Revised ALS Functional Rating Scale (ALSFRS-R), are a proxy for disease progression. However, they are imperfect measures. Individuals are often evaluated at different stages of their disease, making comparisons across patients challenging[4]. While people with ALS invariably decline over time, some of the measures can increase for short durations or reach plateaus[5]. Additionally, the metrics of clinical disease progression are based on subjective assessments of patients' daily functioning, such as the ability to climb stairs 'normally' or 'slowly', which introduces a potential source of error[6]. Furthermore, interventions such as percutaneous endoscopic gastronomy and non-invasive ventilation can affect these clinical metrics. The interconnectedness of function and the variability in the measurement of these clinical metrics present challenges in modeling ALS progression.

The heterogeneity of ALS makes it difficult to determine if a disease-modifying therapy is effectively slowing progression[7,8]. Traditional modeling approaches have dealt with the complexities in ALS by first assuming that ALS outcome measures, particularly the ALSFRS-R, progress in a linear fashion[9–11]. Many ALS clinical trials use changes in the linear slope of ALSFRS-R over time or changes in ALSFRS-R from baseline as primary endpoints[12–14]. For example, edaravone was approved in the USA on the basis of

a 2.5 ALSFRS-R point difference in decline between the treatment and control arms over 6 months (ref. [14]), and the estimated effect from the ALS sodium phenylbutyrate–taurursodiol clinical trial was a change in slope of 0.42 points per month[12]. Large global crowdsourcing analyses designed to produce better models for clinical trials have also assumed a linear decline in ALSFRS-R[15,16].

Despite the widespread use of linear models in predicting patient progression, there is evidence that ALS progression can be nonlinear and can vary across disease severity[17–19]. Several approaches have been proposed to deal with nonlinearity in the context of clinical trials. For example, nonlinear parametric models, which assume a particular shape of the trajectory in advance, have been used to capture these complexities. One notable example is the D50 model, which represents the progression of ALS with a two-parameter sigmoid[20,21]. However, by requiring a particular parametric form, these models are restricted to identifying prespecified trajectory shapes[17,18,22–24], which may not represent the actual heterogeneity in disease progression patterns. Models such as the mixed model for repeated measures can be used in conjunction with unstructured time and covariance structures that reduce reliance on parametric assumptions; however, these models can suffer from being statistically underpowered, especially in clinical cohorts with sparse longitudinal data[25].

Less attention has been paid to a more fundamental question: are there common patterns of clinical progression in ALS? If such patterns exist, they could be used to improve patient stratification, which can impact clinical trial planning[8,26]. Defining distinct clusters can also enable research aiming to identify disease mechanisms that contribute to modulating disease progression in ALS.

[1]Department of Biological Engineering, MIT, Cambridge, MA, USA. [2]Center for Computational Health and MIT–IBM Watson AI Lab, IBM Research, Cambridge, MA, USA. [3]Next Generation Analytics, Palo Alto, CA, USA. [4]Department of Neurology, Emory University School of Medicine, Atlanta, GA, USA. [5]Department of Neurology, Massachusetts General Hospital, Boston, MA, USA. [6]Department of Neurology, Harvard Medical School, Boston, MA, USA. *A list of authors and their affiliations appears at the end of the paper. ✉e-mail: Fraenkel-admin@mit.edu

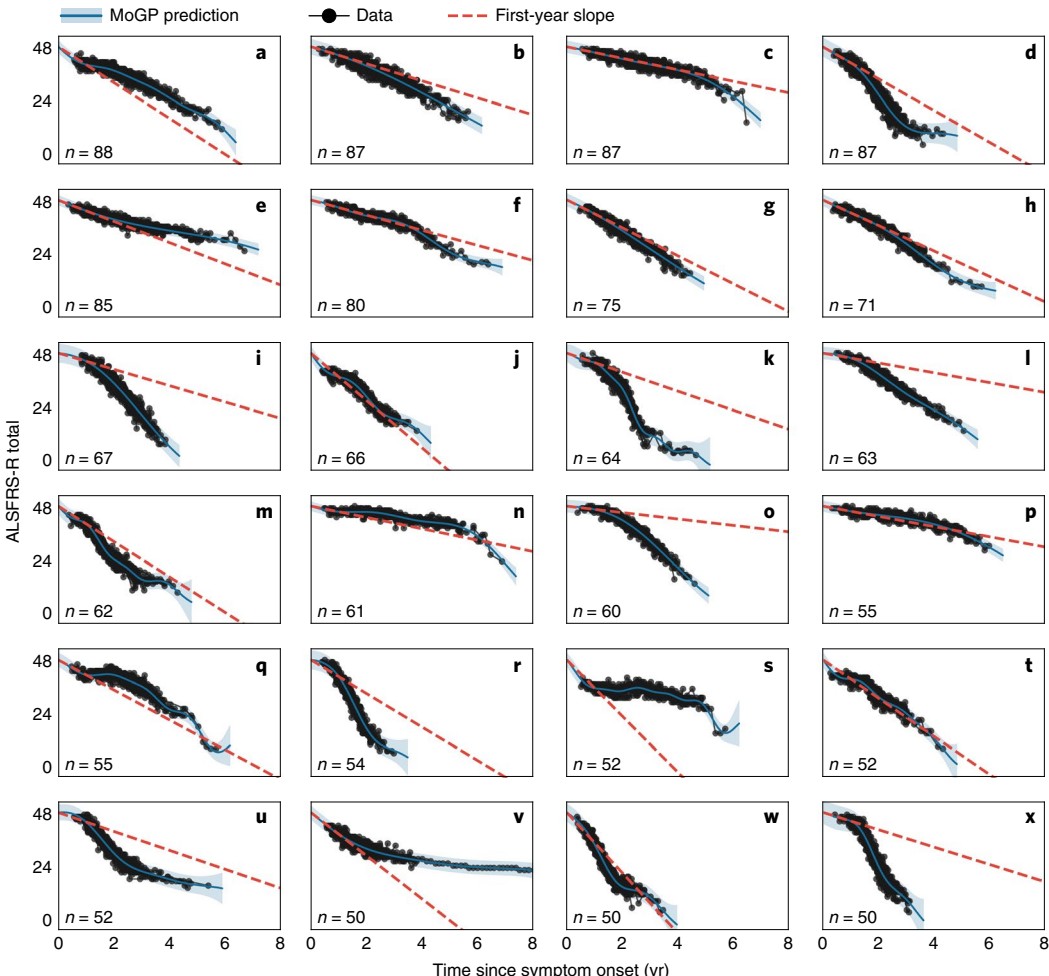

**Fig. 1 | Identifying trajectory clusters with varying patterns of decline, using a mixture of Gaussian processes model.** The 24 largest clusters (out of 92) from PRO-ACT are shown. The first-year slope is calculated as the difference between 48 and the mean cluster score 1 yr after symptom onset, divided by the time from symptom onset. *n* indicates the number of ALS patients in each cluster. The shaded area indicates the 0.95 confidence interval.

Heterogeneity in the clinical progression of ALS may reflect environmental or genetic modifiers of disease, and robustly characterizing heterogeneity in progression patterns can aid in the search for these modifiers. Current clustering efforts for disease progression patterns are limited by requiring parametric assumptions in their cluster assignments or outcome measures[15,19,24]. There is a need for computational methods that can flexibly identify patient clusters with minimal assumptions.

To model the full complexity of ALS progression, we turned to computational methods that are more flexible than traditional parametric models. We propose a framework for aggregating patient trajectories into clusters using Gaussian processes[27,28] and determining the number of clusters using a Dirichlet process mixture model[29,30]. We also modify this mixture of Gaussian processes model to incorporate prior clinical knowledge. For example, since patients with ALS are expected to decline over time, we incorporate monotonic biases into our model, which encourage declining trajectories to be identified but also allow for the detection of patterns that do not fit prior expectations.

We show that this method can improve the characterization of ALS progression patterns, identifying clusters of participants with similar trajectories from longitudinal clinical scores. The nonlinear progression patterns are robust to sparse data and consistent across study populations, and correspond to survival outcomes. While we focus on clinical ALS outcome measures, we also demonstrate

that the method can analyze Alzheimer's and Parkinson's data. Our results provide an advance in modeling progression patterns in ALS and other diseases.

## Results

**Modeling approach.** We developed a mixture of Gaussian processes model with strong inductive bias towards monotonic decline (MoGP) to characterize patterns in disease progression (Extended Data Fig. 1). The model leverages two Bayesian non-parametric methods: Gaussian process regression[27,28] and Dirichlet process clustering[29,30]. Gaussian process regression does not require the specification of a particular functional form, but instead learns trajectories from data, enabling the model to capture a wide variety of possibly nonlinear progression patterns. Dirichlet process clustering can be used to identify clusters from data when it is difficult to specify an expected number of clusters a priori. The use of Dirichlet processes is motivated by the uncertainty in the existence and number of ALS progression subtypes and avoids restrictive modeling assumptions.

Our approach includes notable improvements over previous MoGP models[28,31,32], in that we incorporate clinical knowledge relevant to ALS progression. Specifically, we implement a monotonic inductive bias as well as clinically informed parameter priors for Gaussian process regression and Dirichlet process clustering components. Each component of the model is discussed in more detail in Methods.

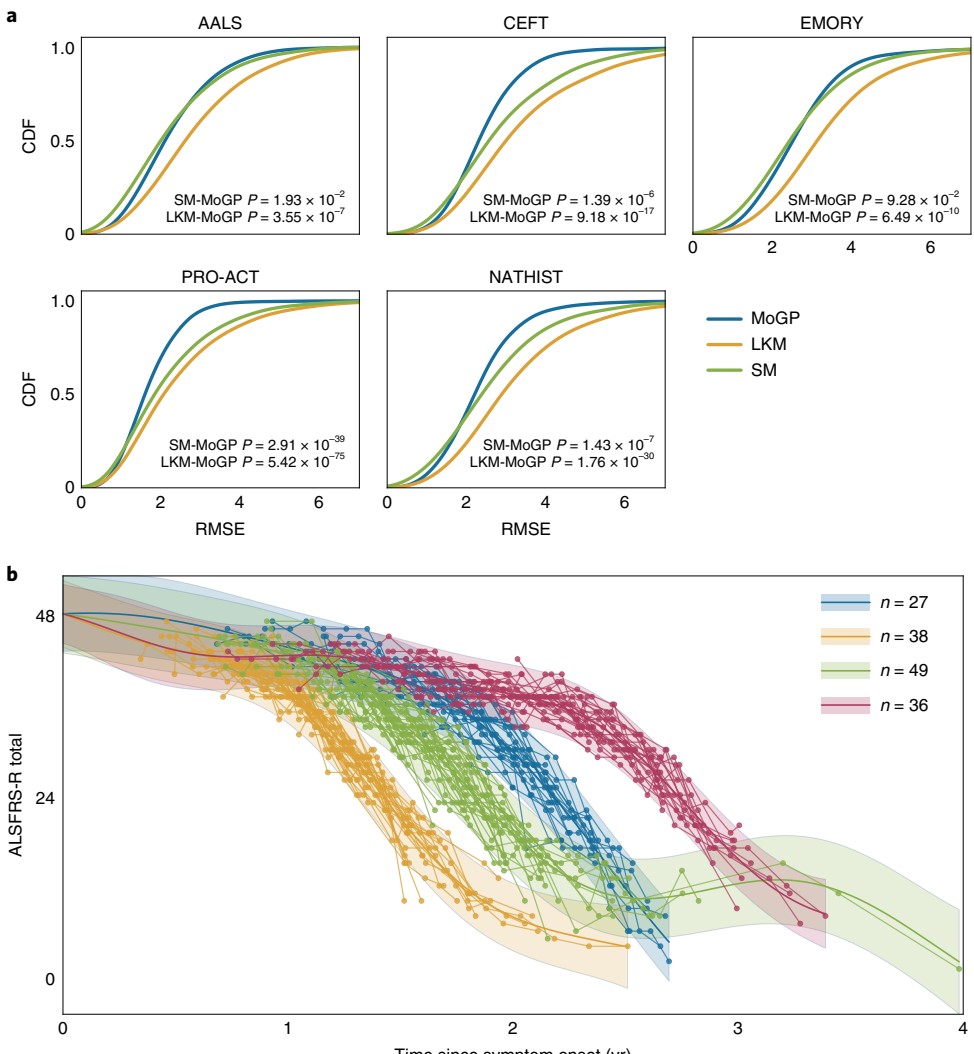

**Fig. 2 | Estimating nonlinearity of trajectories. a**, Cumulative distribution function (CDF) of root mean squared error (RMSE) between a participant's predicted cluster membership and cluster model mean. *P* values calculated with two-sided Kolmogorov–Smirnov two-sample tests between MoGP and LKM distributions, and between MoGP and SM distributions. **b**, A subset of nonlinear clusters from PRO-ACT visualized; *n* indicates the number of ALS patients per cluster. The shaded area indicates the 0.95 confidence interval.

**Elucidating ALS disease progression trajectory patterns.** We sought longitudinal ALSFRS-R scores from a wide range of sources. The model was evaluated on five study populations (Supplementary Tables 1 and 2). Three observational studies were used, Answer ALS (AALS) (https://www.answerals.org/), the Emory ALS Clinic database (EMORY)[3] and the ALS/MND Natural History Consortium database (NATHIST)[33], and two overlapping clinical trial datasets, the Pooled Resource Open-Access ALS Clinical Trials (PRO-ACT)[34,35] and the Clinical Trial of Ceftriaxone in ALS (CEFT)[13,36].

To characterize patterns in ALS progression, we first applied MoGP to PRO-ACT, which is the largest publicly available dataset of ALSFRS-R scores (Fig. 1 and Extended Data Fig. 2). The analysis identified diverse clusters, including some clusters identifying slow-progression populations (Fig. 1n) and others capturing faster-progression groups (Fig. 1r).

Notably, in many cases, the patterns of decline were highly nonlinear, with some following sigmoidal (Fig. 1d,k), convex (Fig. 1m,u,v) and concave (Fig. 1o,q) curves. Linear patterns were also detected in some clusters (Fig. 1g,j,t). To estimate how well a linear model fit in the first year generalizes to subsequent timepoints, we computed the slope of the mean function of each cluster in

the first year after symptom onset ('first-year slope'). This is relevant to previous studies that utilize a first-year slope calculation, including previous ALS DREAM Challenges[15,16]. While this first-year slope closely reflects actual trajectories for the linear clusters (Fig. 1g,j,t), for others it is either an overestimation (Fig. 1i,k,l,o,x) or an underestimation (Fig. 1s,v), indicating nonlinearity in the trajectory pattern. These errors in the first-year estimations can be large; for instance, it overestimates the disease trajectory in cluster K by 24.20 ALSFRS-R points and underestimates the trajectory in cluster V by 9.48 ALSFRS-R points when both are evaluated 3 yr after symptom onset. This diversity highlights the complexity of progression trajectories in ALS. Analysis of other study populations (Extended Data Figs. 3–6) also revealed many clusters that were highly nonlinear.

**Identifying nonlinear patterns across heterogeneous studies.** We compared the performance of MoGP against two other approaches, which assume that progression is linear. A standard model in the field is to calculate linear slopes fit to patient data with an onset anchor[37] (slope model: SM). The slope model is fit to each patient separately and does not identify clusters. We also benchmarked our

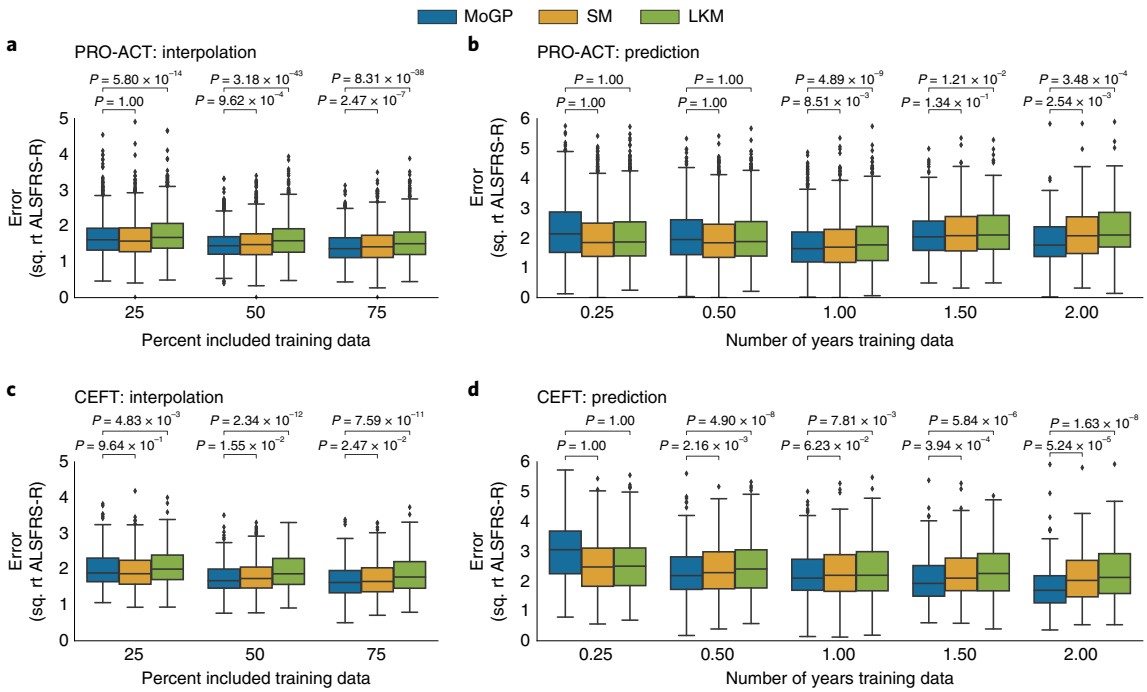

**Fig. 3 | Evaluating robustness of cluster assignments with sparse datasets. a,c**, MoGP, LKM and SM were trained on interpolated data and RMSE was calculated between withheld data and the mean predicted trajectory. **b,d**, Models were trained on right-censored data. *P* values were calculated with a Wilcoxon signed-rank one-sided test. The box plot represents the interquartile range around the mean; whiskers indicate the proportion (1.5) of the interquartile range past the low and high quartiles to extend the plot whiskers. Points outside the whisker range represent outlier samples. Number of patients evaluated: **a**, n = 1,327 patients for all comparisons; **b**, 0.25 yr, n = 2,786; 0.5 yr, n = 2,465; 1 yr, n = 1,379; 1.5 yr, n = 261; 2 yr, n = 135; **c**, n = 228 for all comparisons; **d**, 0.25 yr, n = 453; 0.5 yr, n = 437; 1 yr, n = 323; 1.5 yr, n = 215; 2 yr, n = 130.

model against a mixture of Gaussian processes model with a linear kernel (linear kernel model: LKM). The LKM retains the ability to cluster trajectories using a Dirichlet process but does not allow for nonlinear functions, allowing us to separate the contribution of clustering and the assumption of linearity in our models.

For all study populations, the error was lower in the MoGP model than in the LKM and SM (Fig. 2a). Across the populations, using the MoGP reduced error by more than one ALSFRS-R point as compared with the LKM for at least 27.16% of participants; at least 8.33% of patients have an improvement in accuracy greater than two ALSFRS-R points (Supplementary Table 3). Importantly, the error of the MoGP was lower even though the LKM used a larger number of clusters to model the data (Supplementary Table 6). It is also notable that the MoGP, which identified clusters as large as 88 participants, was able to match or outperform the patient-specific SM (Fig. 2a and Supplementary Table 4), which would have been expected to markedly outperform MoGP if substantial nonlinear structure did not exist in the data. The results are replicated across the five different datasets, suggesting that complex nonlinearity is a common feature of ALS progression and is not a unique feature of a single dataset.

The clusters with the most substantial nonlinearity often followed sigmoidal trajectory patterns, with varying inflection points (Fig. 2b). In some of these clusters, patients had slow progression for a period of time, followed by a consistent sharp decline. This pattern of progression appears consistent with a sudden loss of ability to carry out functions that we refer to as a 'functional cliff'. In other cases, the pattern is more consistent with a rapid period of decline followed by a slower phase. Since there are many settings in which patient-specific parametric models are very useful, we compared our model with a patient-specific sigmoidal model (SG)[20]. Somewhat surprisingly, despite the fact that the MoGP

models groups of patients, rather than individuals, MoGP outperforms a patient-specific sigmoid model by one or more ALSFRS-R points for 4.20–9.43% of patients across the studies (Supplementary Table 5). This indicates that, while a sigmoidal model captures much of the nonlinearity, it does not represent the full complexity of progression patterns.

MoGP clusters varied considerably in their rates of progression and the stability of their progression patterns. MoGP enables the characterization of each of these properties through the mean function slope and kernel function length-scale parameters respectively, both of which are learned and optimized through the training process. The model provides estimates for each of these parameters, and these can be used to approximate similarity between clusters depending on the desired clustering property (Extended Data Fig. 7).

Clustering trajectories on the basis of the optimized slope and length-scale parameters reveals interesting patterns (Extended Data Fig. 7). The dominant clinical progression patterns in ALS are sigmoidal fast progression (Extended Data Fig. 7b, 17.48% of individuals), stable slow progression (Extended Data Fig. 7e, 17.38%), unstable slow progression (Extended Data Fig. 7f, 32.98%) and unstable medium progression (Extended Data Fig. 7d, 30.82%). As might have been expected, some types of progression were associated with specific sites of onset. Clusters with fast sigmoidal progression have the highest percentage of individuals with bulbar onset (30.14% of individuals), while those with stable slow progression have the highest percentage of individuals with limb onset (76.97%) (Supplementary Table 8).

Overall, the MoGP model promotes the ability to learn these complex disease progression trajectories better than currently used clinical models, while stratifying patients to reveal common patterns of disease.

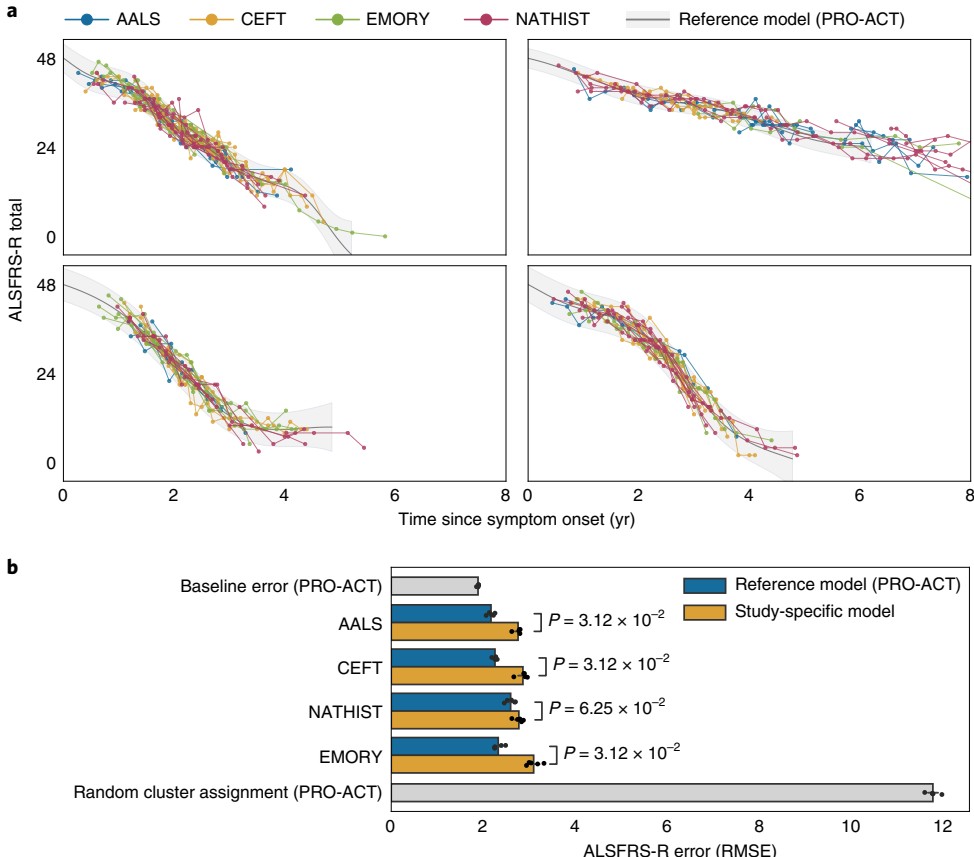

**Fig. 4 | Assessing trajectory consistency across datasets. a**, The reference model was trained on PRO-ACT and used to predict progression trajectories of participants in other datasets; the four largest reference model clusters are shown. **b**, Average test error between cluster mean function and participant ALSFRS-R scores, using the reference model and study-specific models. *P* values were calculated with a Wilcoxon signed-rank one-sided test. The error bars show the 0.95 confidence interval around the mean. *N* = 5 test–train splits.

**Evaluating the robustness of the clusters to sparse data.** As clinical data for ALS patients are often incomplete or sparse, we sought to evaluate MoGP performance in these settings. We tested robustness using PRO-ACT, which is the largest of our sources and is a compendium of data from several clinical trials. We also tested robustness using data from CEFT, which is a small clinical cohort within PRO-ACT that may be more representative of common clinical settings. We compared MoGP's performance against LKM and SM.

We first evaluated the model's ability to recreate randomly withheld data points ('interpolation'). Across all interpolated tests for PRO-ACT, we found that the clusters identified by MoGP had lower reconstruction error than the LKM (Fig. 3a, $P \leq 1 \times 10^{-4}$), and a lower error than the SM when 50% and 75% of training data are included (Fig. 3a, $P \leq 1 \times 10^{-3}$). These trends persisted when compared with CEFT (Fig. 3c).

One of the most common uses for trajectory modeling is to predict future ALSFRS-R scores. We therefore evaluated the model's ability to predict future ALSFRS-R scores for patients with right-censored data ('prediction'). In clinical trials, these predictions are often made with the SM. For PRO-ACT, when only three or six months of data from baseline were provided, the SM and LMK were the most accurate (Fig. 3b). However, when one or more years of training data were provided, the MoGP model outperformed the LKM and SM (Fig. 3b, $P \leq 1 \times 10^{-2}$, except for 1.5 yr, where $P = 1.34 \times 10^{-1}$ for SM), and more accurately predicted future disease progression by more than 0.22, 0.41 and 1.28 ALSFRS-R points at 1, 1.5 and 2 yr respectively. This trend was strengthened in CEFT, in which six

months of training data were sufficient to see an improvement in progression forecasting (Fig. 3d, $P \leq 1 \times 10^{-1}$).

For the majority of comparisons, the MoGP identified fewer clusters per mixture model than the SM or LKM, indicating that the lower reconstruction error was not due to overfitting of the cluster assignments (Supplementary Fig. 1).

**Transferring trajectories across study populations.** Because ALS is heterogeneous and characteristics of study populations can differ considerably, it is important to test whether trajectory models capture patterns that are consistent across populations. To answer this question, we trained MoGP on a large database ('reference model') and used it to predict patient trajectories in other study populations that varied in data collection frequency and follow-up period. We benchmarked the MoGP results against models in which both the test and training sets were derived from the same study population ('study-specific models'). The study-specific models allow us to evaluate possible overfitting of the reference model. If the reference model was overfit, we would expect it to have a much higher error than the study-specific models.

We found that the reference model, trained on PRO-ACT, demonstrated strong performance on external datasets, indicating that the trajectory clusters are not overfit to the reference model data (Fig. 4a). Importantly, we found that for all test datasets the reference model outperformed the study-specific models (Fig. 4b, $P = 0.0312$ for AALS, CEFT, EMORY; $P = 0.0625$ for NATHIST). AALS had the lowest error when the reference model was used (2.16 ALSFRS-R

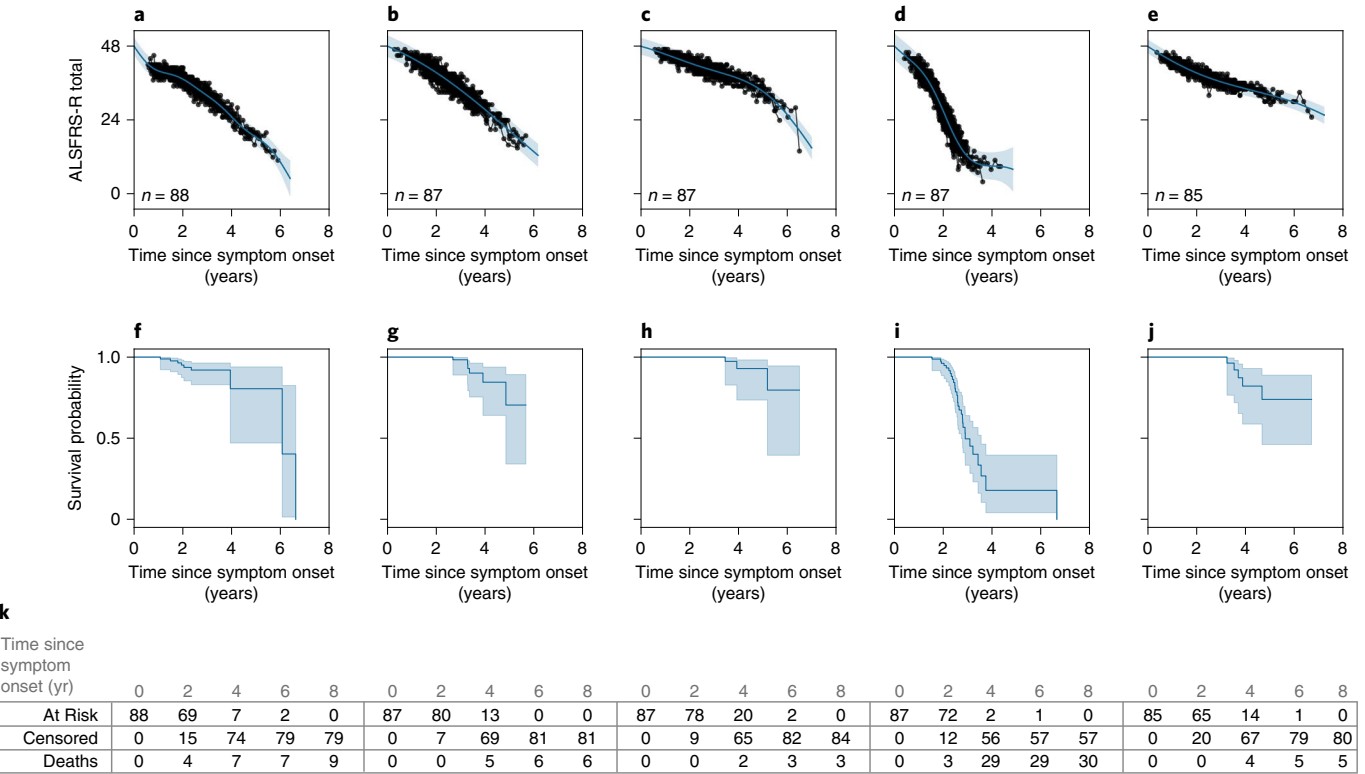

**Fig. 5 | Survival outcomes for trajectory clusters. a–k,** The five largest PRO-ACT clusters are shown with MoGP clusters (**a–e**) and associated Kaplan–Meier survival curves (**f–j**). *n* indicates the number of ALS patients in each cluster. The shaded area indicates the 0.95 confidence interval. **k,** The number of individuals at risk, censored and with recorded deaths observed at each time displayed.

points), followed by CEFT (2.25), EMORY (2.32) and NATHIST (2.59) (Fig. 4b). These errors were similar to the baseline error (1.88 ALSFRS-R points) when the reference model was tested on the held-out data from PRO-ACT, the study on which it was trained. We additionally benchmarked the model against a reference model for which the cluster labels were randomly shuffled. This randomized control had a mean error of 11.74 ALSFRS-R points, which was much higher than the errors of the reference models. Given that CEFT is a subset of PRO-ACT, it is interesting that the CEFT study-specific model had a higher error than its reference model counterpart; these results suggest that the larger size of the PRO-ACT dataset may allow it to capture trajectories more accurately. The reference model's ability to outperform all of the study-specific models is strong evidence that the trajectory patterns identified by MoGP are transferable across ALS study populations.

**Corresponding survival outcomes with trajectory clusters.** Next, we evaluated if the MoGP clusters, which were trained only on ALSFRS-R data, were able to reflect the duration of patient survival from symptom onset to death. The results of the Kaplan–Meier analysis are presented in Fig. 5. Some clusters (Fig. 5c,e) reflected longer survival durations, with very few deaths recorded, while other clusters reflected shorter durations. For example, cluster D had a median survival of 2.90 yr from symptom onset, corresponding to faster progression (Fig. 5d,i). Of all pairwise combinations of clusters, 63.40% corresponded to differential survival outcomes when MoGP was used. By contrast, when LKM was used 50.99% of pairwise combinations of clusters corresponded to differential survival outcomes ($P < 0.05$). These results demonstrate that incorporating nonlinearity improves the correspondence of clusters to survival outcomes and provides evidence that these progression clusters are clinically relevant.

**Characterizing patterns of decline in alternative ALS measures.** In addition to ALSFRS-R scores, there are other important clinical metrics that can be used to monitor ALS disease progression. One is forced vital capacity, which is a spirometer-based measure of lung function and has been used as an indicator of survival and disease progression[38]. Furthermore, while the ALSFRS-R total is commonly used as an aggregate measure, its component subscores measuring fine motor, gross motor, bulbar and respiratory function can also be analyzed to identify subscore-specific patterns. When we applied MoGP to forced vital capacity and ALSFRS-R subscores from PRO-ACT, we saw that the nonlinearity persisted in these domains. The nonlinear trajectories were particularly pronounced for forced vital capacity and bulbar function (Fig. 6).

A key advance of this work is the identification of clusters of patients, which can be used to investigate genetic or environmental causes that may underlie ALS progression. For instance, the *C9orf72* repeat expansion is the most common of the known causes of ALS, and it is associated with faster-progression ALS, as indicated by reduced survival[39,40]. However, even among patients who share this common genetic cause of ALS, there is some evidence of uncharacterized heterogeneity in ALS progression patterns[41]. As an example use-case, we asked if MoGP can be used to stratify patients who carry this repeat expansion. We analyzed data from AALS, a study population with both clinical and molecular data available. The patients with the *C9orf72* repeat expansion did not correspond to a single cluster, supporting the hypothesis of heterogeneous progression within this group. As more data accumulate, such analyses could aid in the search for genetic or environmental variables that modify the aggressiveness of *C9orf72*.

**Revealing patterns in Alzheimer's and Parkinson's endpoints.** The MoGP approach can be applied to functional rating scales that

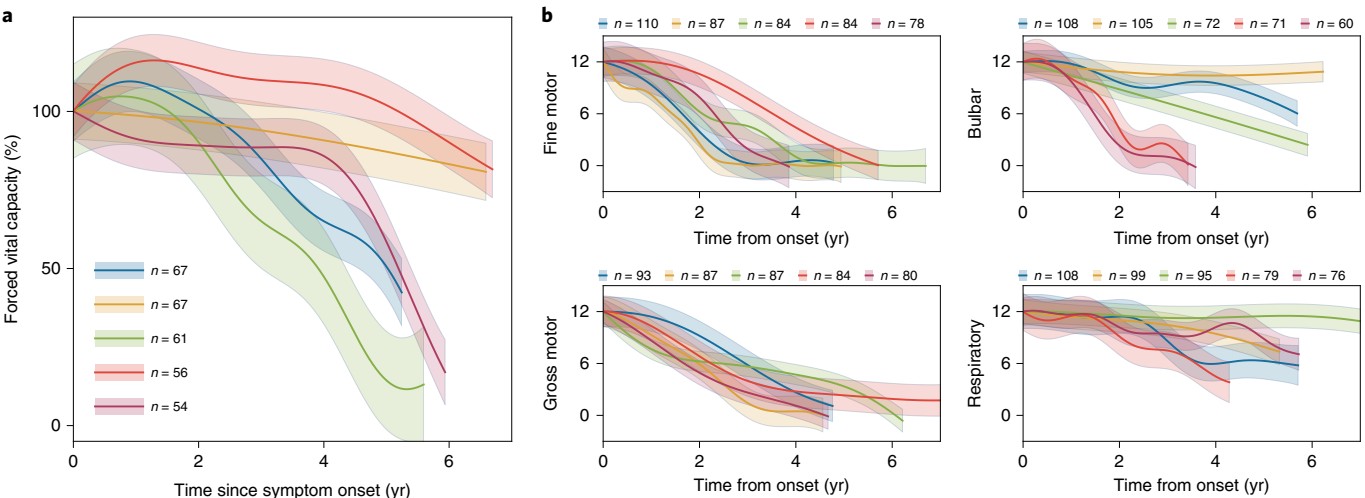

**Fig. 6 | MoGP trajectory patterns for secondary endpoints of ALS disease progression.** Measures include **a**, forced vital capacity and **b**, ALSFRS-R subscores (fine motor, gross motor, bulbar, respiratory domains). Trajectory colors for each panel are unrelated and correspond to the relative number of participants in each cluster, scaled from the largest to the smallest of the five largest clusters from PRO-ACT. Participants with minimal change in score (≤1 point) were excluded from the model. *n* indicates the number of ALS patients in each cluster.

are widely used in other neurodegenerative diseases. We applied MoGP to the Alzheimer's Disease Assessment Scale—Cognitive Subscale (ADAS-Cog-13 (refs. [42,43]). The model showed a range of disease progression patterns, with varying severities of progression (Extended Data Fig. 8a and Supplementary Fig. 5). The majority of the largest clusters showed linear trajectories, in which the first-year slope appropriately captured later progression; clusters E and H, while largely linear, deviate from the first-year slope, showing counterexamples to this trend. It is noteworthy that the clusters varied substantially in the rates of conversion of mild cognitive impairment (MCI) to Alzheimer's disease. Ninety percent of those in cluster F had an MCI diagnosis at baseline, compared with 5.26% in cluster G (Supplementary Table 9).

Similarly to ALS and Alzheimer's disease, Parkinson's disease is heterogeneous in its symptom presentation and progression, which creates challenges in therapeutic discovery. Unlike ALS and Alzheimer's disease, there are widely used medications for Parkinson's disease that can provide symptomatic relief, although they do not slow or stop the progression of the disease[44]. We characterized patterns in motor decline by applying MoGP to Part III of the Movement Disorder Society—Unified Rating Scale (MDS-UPDRS)[45] using only data from the 'off state', that is, when not affected by medications. MoGP identified a number of progression trajectories (Extended Data Fig. 8b and Supplementary Fig. 6), with some showing stability of motor scores (clusters C, F), while others showed clear motor function decline (clusters A, B, D).

Interesting trends emerged from this analysis. Over 90% of individuals in clusters with an unstable slow progression pattern (Supplementary Fig. 8 and Supplementary Table 10) had tremor-dominant (TD)[46] Parkinson's disease, as opposed to postural instability/gait difficulty (PIGD). In contrast with previous studies of the linearity of MDS-UPDRS scores[47], our results also point to nonlinear complexity in some clusters (clusters C, E, G) (Extended Data Fig. 8b). These analyses demonstrate that MoGP's flexibility enables it to characterize long-term heterogeneity in time-series metrics in a number of diverse clinical settings.

## Discussion
The improved performance of an MoGP model over the slope and linear kernel models indicates that linear models are insufficient to capture the heterogeneity in ALS disease progression. While some

patients do indeed have linear trajectories, a substantial portion of patients have nonlinear trajectories. Our work also finds that, while a simple parametric nonlinear model—a two-parameter sigmoid— is better than a linear model, it still fails to capture the full range of patient trajectory patterns, motivating the use of non-parametric models that can capture both linear and nonlinear trajectories.

Previous work has suggested that the functional cliff patterns seen here may be a result of inconsistencies in the ALSFRS-R or issues related to the ordinal scale used in ALSFRS-R as opposed to a linearly weighted interval scale[48,49]. However, the consistency of MoGP-identified patterns across different study populations suggests that the patterns are not the result of deficiencies in the ALSFRS-R. Critically, we also observed nonlinear patterns in vital capacity scores, which are measured independently of ALSFRS-R scores. These findings support the view that nonlinearity is reflective of changes in patient function and not problems in measurement. These findings also have implications for the analysis of clinical trials, many of which use ALSFRS-R and vital capacity metrics as primary or secondary endpoints. In many trajectory clusters, functional cliffs or sigmoidal patterns in disease progression may obscure the detection of therapeutic efficacy if linear models are used. Our results support accounting for nonlinearity when evaluating ALS clinical trial efficacy, with particular salience for clinical trials that are 1 yr in duration or longer.

Our work also demonstrates how existing clinical databases in ALS can be leveraged to enable the characterization of disease progression in sparse datasets from different study populations. A MoGP model trained on the PRO-ACT database accurately predicted trajectories for clinical datasets from AALS, EMORY, CEFT and NATHIST datasets. The transferability of MoGP-identified clusters across the datasets indicates that the trajectory cluster patterns are robust to batch effects due to clinician or site differences, and may reflect underlying disease processes. One of the properties of the Dirichlet process model underlying MoGP clustering is that it will naturally scale the number of identified clusters within a given dataset depending on the number of samples in that dataset; we can use a reference model on a clinical cohort of any size. This non-parametric property of the model underlies the difference in the total number of clusters found in the varying datasets. Conversely, when it is useful to analyze fewer clusters the trajectories can easily be grouped together on the basis of their mean slope

and length scale, revealing dominant modes of disease progression. The identification of these clusters creates an opportunity to search for molecular, environmental or other factors that may modify disease progression.

As in many clinical studies, the datasets and therefore the progression patterns in this analysis are influenced by both selection bias and attrition bias. Selection bias refers to the sample of the population that is included in each study. Studies such as AALS, which require enrollment and consent to undergo additional monitoring, tend to be biased towards slower-progressing ALS. The EMORY dataset, which has a high percentage of enrollment from the clinic, is likely to be more reflective of a clinical population, although it reflects a group of patients with higher rates of progression on average. Overall though, observational studies tend to have less standardized frequencies of data collection and sparser measurements. On the flip side, clinical trial datasets typically collect extensive longitudinal data, but because of enrollment criteria can be skewed towards faster-progression individuals. The variation in ages of onset and prevalence of sites of onset differ across clinical cohorts, which can indicate additional potential selection biases. Other variables that can be used to evaluate selection bias but were partially missing or unavailable across our studies include diagnostic delay, forced vital capacity, frontotemporal dementia and *C9orf72* status. Attrition bias also plays a strong role in ALS datasets, given the rapid pace of disease progression, with patient monitoring becoming increasingly difficult in late-stage disease; this bias may particularly affect the tail end of the identified trajectory patterns. Given the large sample size in our study, and the consistency of the patterns across datasets, we expect that we are sampling the clinical population as broadly as possible, although future work will involve determining the extent to which these trajectories remain consistent in new datasets.

Ultimately, by identifying clusters of patients who have similar disease progression trajectories, these models could be used to identify molecular correlates that may be associated with ALS progression subtypes. While this work focuses on ALSFRS-R and vital capacity, the field of ALS has identified a growing number of molecular biomarkers and clinical metrics in which progression is poorly understood[50,51]. This paper points to the complexity of disease progression in ALS and the necessity of more accurately accounting for heterogeneous trajectory patterns in clinical trial models and research studies.

## Methods

**Study populations.** ALS data in this study were collected from five cohorts: PRO-ACT, CEFT, AALS, EMORY and NATHIST (Supplementary Tables 1 and 2). All scores used for this analysis are clinician reported. The populations varied in size, with PRO-ACT having the largest total number of participants (2,923 participants with at least three ALSFRS-R visits recorded). The populations differed in the median number of months followed (between 11 and 17 months) and the median frequency of clinical visits (between four and nine visits). The median slope between the populations also varied, with CEFT and EMORY having the fastest-progression populations (−0.84 and −0.89 ALSFRS-R points/month, respectively), and AALS having the slowest-progression population (−0.55 ALSFRS-R points/month). CEFT had a median of 16.80 months of follow-up, while PRO-ACT had a median of 11.95 months, indicating that CEFT participants likely comprise some of the longest subject records in PRO-ACT. The differences between the populations allowed us to measure the robustness of our model to data collection methods, frequency of clinical visits and duration of follow-up.

**Modeling approach.** We characterize disease progression in ALS using a framework for identifying trajectory patterns from longitudinal data. While previous work on disease progression modeling has focused on patient-specific prediction models[16,52,53], a critical advance of this work is the characterization of distinct and large trajectory clusters. Furthermore, we provide a principled approach to characterizing the shapes of disease progression patterns in ALS, which leverages Bayesian non-parametric methods to minimize the number of assumptions that are required for regression models. We show that this method can flexibly be applied to a number of functional clinical measures for progressive diseases. Each component of the model is detailed further below. Further

details, including the mathematical specification of the model, can be found in Supplementary notes.

The modeling approach of clustering over temporal progression patterns has been shown to improve the characterization of disease progression in other conditions. For example, Peterson et al. demonstrated the use of an autoregressive Gaussian process model for predicting metrics of Alzheimer's progression[54]; however, the model made a fundamentally different assumption about the structure of the data—that there is a single global progression type, and that each patient follows a noisy version of this global progression type—which is an assumption that does not capture the full heterogeneity of ALS phenotypes. Furthermore, the model requires fixed time intervals of visits, which are not available in many clinical ALS datasets[54]. Zhao et al. present a related clustering approach in multiple sclerosis, although their model relies heavily on prior domain knowledge on how to group patients into subgroups, which has not as yet been clearly defined in ALS[55]. Other models, such as additive Gaussian process regression[56], can be used to characterize patterns in time-series data, although they lack the ability to stratify patients into disease subtypes.

*Gaussian process regression.* Gaussian process regression allows the identification of nonlinear trajectory patterns while making minimal assumptions about the shape of the trajectory functions[27,28]. A Gaussian process is specified by a mean function and a covariance kernel. Because we expect ALS trajectories to be smooth functions with no discontinuities, our MoGP model uses a squared exponential kernel. The squared exponential kernel has two parameters: the signal variance, which determines the average distance of the function from the mean, and the length scale, which specifies the smoothness of the function. Each of these parameters is determined during the learning phase using the training data.

*Monotonic inductive bias.* Because ALS trajectories are expected to decline over time, we use a negative linear function in the Gaussian process models of MoGP. To further encourage declining trajectories, we modify the Dirichlet process clustering algorithm, such that an individual can only be placed in a cluster if their score at their initial visit is not substantially higher than the mean function of the current cluster at that point. We also impute an onset-anchor value, a maximum score of a clinical metric assigned to the date corresponding to symptom onset, which has been previously shown to improve prediction in ALS trajectories[37].

*Dirichlet process clustering.* Dirichlet process mixtures[29,30] can be used to identify clusters in data without needing to specify an expected number of clusters in advance. This unsupervised learning model begins by assuming that an infinite number of clusters can exist, and then narrows its prediction to a limited number of components best supported by the observed data. In our case, each mixture component is a function drawn from a Gaussian process. The resulting model clusters patient trajectories by probabilistically assigning them to those components that best explain them. The number of patients in each cluster is also learned from the model, and clusters can differ in size from each other. Through this data-driven approach, the algorithm can learn clusters of ALS patients who share disease progression patterns. The method can also predict the cluster membership and the disease progression pattern of a participant not included in the model, and provide an estimate of the confidence of this prediction.

**Model evaluation.** *Evaluating trajectory nonlinearity.* We evaluated how generalizable a linear model trained in the first year of disease progression is to subsequent data points, by calculating an anchored first-year slope. This was computed as the following: (48 − cluster mean function at 1 yr)/time from symptom onset. Anchoring indicates that a score of 48 (the maximum of the ALSFRS-R scale) is imputed at the time of symptom onset[37]. We compared our MoGP model against two benchmark linear models: an anchored slope model (SM), which is patient specific, and a mixture of Gaussian processes model with a linear kernel (LKM), which clusters patients using a linear parametric model. Additionally, to evaluate the extent to which a nonlinear parametric model represents ALS progression, we compared our model against a patient-specific two-parameter sigmoidal model (SG)[20].

Because we are proposing the use of a data-driven model, we aimed to be as conservative as possible in removing patients from the dataset so as to not introduce additional selection bias. For this analysis, participants were excluded from the model if fewer than three complete ALSFRS-R visits were recorded, the first visit was more than 7 yr from symptom onset or an increase of more than six points in ALSFRS-R between consecutive visits was recorded (Supplementary Table 1). The ALSFRS-R is an updated version of the previously used ALSFRS metric, and includes additional questions measuring dyspnea, orthopnea and respiratory insufficiency[6]. The ALSFRS-R measure was used here because it is the current standard in clinical trial analysis[12–14]. Seven years was selected as the point at which longitudinal data became sparse. Six ALSFRS-R points was selected because a jump such as this was unlikely to be seen unless there was a data-entry error.

For each model, the RMSE between a participant's measured scores and their predicted cluster mean function were calculated. The RMSE was compared between the models; a lower RMSE indicates reduced error in that model and better model performance.

*Robustness to sparse data.* We simulated sparsity by withholding data and assessed the model's ability to perform two tasks: (1) interpolation of ALSFRS-R scores for a patient with randomly withheld data points, and (2) forecasting future ALSFRS-R disease progression for patients with right-censored data. We tested this using PRO-ACT and CEFT. To have sufficient longitudinal measurements, for interpolation experiments we only included participants with ten or more longitudinal ALSFRS-R visits, and for prediction experiments we only included participants with four or more visits (Supplementary Table 7).

The reconstruction error for each participant was calculated using the RMSE between the original withheld data points and predicted values from the mean function for the participant's trajectory cluster. This was done across all interpolated tests, in which 25%, 50% and 75% of clinic visits per patient were provided as training data, with selections randomly interspersed across visits. We additionally evaluated the ability of MoGP to predict future progression by using right-censored data with varying numbers of training data (including visits within 0.25, 0.5, 1, 1.5 and 2 yr since baseline visit).

*Model generalizability.* To evaluate whether clusters derived from one study population could be used to model external study populations, we trained a reference model and evaluated the transferability of this model to unseen ALS patient data. We predicted the cluster membership for each participant, and calculated the RMSE between the participant ALSFRS-R scores and the mean function of their predicted cluster.

We split all of our study populations into test and training datasets (60% train, 40% test; repeated across five randomly split test–train datasets). For our reference MoGP model, we used the training data from PRO-ACT, which was chosen because it contained the largest number of samples and is publicly available. For AALS, EMORY, CEFT and NATHIST, we used the training data from each study to train a separate model (study-specific model). For each study's remaining test data, we predicted the trajectory function using the reference model and the study-specific model. To approximate the minimum error expected, we calculate the reconstruction error when the reference model is applied to the test set from PRO-ACT, the same study on which it is trained ('baseline error'). We also benchmark against the error when cluster labels on the reference model are randomly shuffled ('random cluster assignment').

*Relationship to alternative outcome measures.* We calculated the Kaplan–Meier survival probability curves for the largest MoGP clusters identified from PRO-ACT. If no death was recorded, the participant was marked as censored using the latest date of a recorded ALSFRS-R score.

We trained MoGP models on forced vital capacity percentages (calculated as the maximum of three trials) and ALSFRS-R subscores (fine motor, gross motor, respiratory and bulbar domains). A maximum score of 100% was used for the forced vital capacity percentage model, and a maximum score of 12 was used for ALSFRS-R subscores.

We applied our method to ADAS-Cog-13 (refs. [42,43]) from the Alzheimer's Disease Neuroimaging Initiative (ADNI)[57]. Individuals with a confirmed Alzheimer's disease diagnosis at any point of the data collection were included in the model; this also included individuals who began the study with MCI and then converted to an Alzheimer's disease diagnosis. To ensure sufficient longitudinal data, individuals with fewer than three longitudinal visits were excluded, with a total of 331 individuals included in the model. The correlation between the learned clusters and MCI to Alzheimer's disease conversion was then calculated.

We also applied our method to the MDS-UPDRS[45] scale from the Parkinson's Progression Markers Initiative dataset[58]. In contrast to ALS and AD, for Parkinson's disease there are medications that can mitigate symptoms although not long-term progression of the disease[44]. Because we were interested in characterizing progression patterns when not affected by medications, we focused on measurements of the MDS-UPDRS Part III in the off state, which is defined as either before the initiation of medication or after abstaining from medication for at least 12 h. Individuals with fewer than three longitudinal off-medication scores or a first visit more than 10 yr from symptom onset were excluded, with a total of 397 individuals included in the model. We calculated the correlation between the clusters and Parkinson's disease subtypes of PIGD and TD, with the designation of PIGD/TD calculated following the method previously described by Stebbins et al.[46]. For the purpose of analyzing the Parkinson's disease subtype correlation with cluster membership, we focused on individuals with a stable PIGD/TD designation (one that does not change over the course of the disease).

*Statistics and reproducibility.* To compare the cumulative distribution function of the RMSE between a participant's predicted cluster membership and cluster model mean, *P* values were calculated with Kolmogorov–Smirnov two-sample two-sided tests. For interpolation and prediction experiments, to determine if a model error had decreased between the LKM or SM and the MoGP, a Wilcoxon signed-rank one-sided test was used. To assess trajectory consistency between reference models and study-specific models, a Wilcoxon signed-rank one-sided test was used. To calculate survival curves, a Kaplan–Meier estimator was used, with *P* values calculated via the logrank test with FDR correction. *P* values for cluster correlations were calculated using a hypergeometric test.

**Reporting summary.** Further information on research design is available in the Nature Research Reporting Summary linked to this article.

## Data availability

A pretrained reference model for this study can be downloaded here: http://fraenkel.mit.edu/mogp
Source Data for Figs. 2–4 and Extended Data Fig. 7 are available with this manuscript. Source Data for Fig. 1 and Extended Data Fig. 2 are available as a Python object from http://fraenkel.mit.edu/mogp. Other source data are unavailable at this time because they contain patient-level clinical data; however, all figures can be generated using the code provided, after downloading the datasets listed below.
Clinical data for this study can be obtained from the following sources. AALS (ClinicalTrials.gov identifier NCT02574390) is available for download in the Answer ALS data portal (data.answerals.org). PRO-ACT can be downloaded from the PRO-ACT database (https://nctu.partners.org/ProACT). CEFT (ClinicalTrials.gov identifier NCT00349622) can be downloaded from National Institute of Neurological Disorders and Stroke (NINDS) (https://www.ninds.nih.gov/Current-Research/Research-Funded-NINDS/Clinical-Research/Archived-Clinical-Research-Datasets). EMORY is restricted access at this time due to containing information that could compromise patient privacy, but available with permission from Dr. Jonathan Glass (jglas03@emory.edu) for legitimate research. Response to requests will be provided within two weeks, all data provided will be fully de-identified, a DUA will need to be established and the source data will need to be acknowledged in any publications. NATHIST is available from the ALS/MND Natural History Consortium (https://www.data4cures.org/requestingdata) with a summary of proposed data use, data elements requested and publication intent. The Parkinson's Progression Markers Initiative can be downloaded, with a data use agreement, online application and compliance with publication policy (https://www.ppmi-info.org/access-data-specimens/download-data). Applications for data access are reviewed by the Data and Publications Committee within one week of receipt. ADNI can be downloaded through the LONI Image and Data Archive (https://adni.loni.usc.edu/data-samples/access-data/#access_data). Access is contingent on adherence to the ADNI Data Use Agreement and its publication policies. The application process includes the acceptance of a data use agreement and submission of an online application form. The application must include the investigator's institutional affiliation and the proposed uses of the ADNI data. ADNI data may not be used for commercial products or redistributed in any way.

## Code availability

We provide the Python code for the MoGP framework as well as a pretrained reference model that researchers can use to generate predictions of cluster membership and trajectory function from input patient data. We also provide a pip-installable Python package associated with this work (mogp). All code used for data processing, modeling and figure generation can be found at https://github.com/fraenkel-lab/mogp. Code is also deposited on Zenodo (license BSD 3-Clause; https://doi.org/10.5281/zenodo.6744399)[59].

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

## Acknowledgements

Data used in the preparation of this article were obtained from the PRO-ACT database, the ALS/MND Natural History Consortium, the Parkinson's Progression Markers Initiative database and the ADNI database. This research includes the National Institute of Neurologic Disease and Stroke's Archived Clinical Research data (Clinical Trial of Ceftriaxone in ALS, M. Cudkowicz, Massachusetts General Hospital) obtained from the NINDS Archived Clinical Research Datasets webpage. Additional information about the studies can be found in Supplementary Acknowledgements. The Answer ALS organization, ALS Finding a Cure and Packard Foundation supported the collection of the Answer ALS clinical dataset used in the manuscript. The Muscular Dystrophy Association contributed funding to the Emory ALS Clinic database that was included in this research. C.N.F. received funding from the Department of Veterans Affairs of Research and Development (IK2CX001595-02) and the Department of Defense (AL200156). K. Sachs received funding from the Muscular Dystrophy Association (award 574137). D.R. received funding from the NSF Gradate Research Fellowship Program (GRFP) and Siebel Scholars Fellowship. E.F. and D.R. received funding from Answer ALS, MIT–IBM Watson AI Lab (W1771646), the United States Army Medical Research Acquisition Activity (W81XWH-21-1-0245) and NIH (U54NS091046). T.M.H. received funding from the NIH/NINDS (K23NS099380). None of the organizations had any influence on the writing of the manuscript or the decision to submit it for publication.

## Author contributions

D.R., K. Severson and S.G. contributed to model development and analyzed data. D.R., K. Severson, K.N. and E.F. contributed to project design. D.R. wrote the manuscript with input and revisions from all authors.

**Competing interests**

K.N., K. Severson and S.G. were employed by IBM Research during this project. K. Sachs consults for Modulo Bio Inc.

**Additional information**

**Extended data** is available for this paper at https://doi.org/10.1038/s43588-022-00299-w.

**Correspondence and requests for materials** should be addressed to Ernest Fraenkel.

**Peer review information** *Nature Computational Science* thanks Mamede de Carvalho, Cassie S. Mitchell and Henk-Jan Westeneng for their contribution to the peer review of this work. This article has been peer reviewed as part of Springer Nature's Guided Open Access initiative. Primary Handling Editor: Ananya Rastogi, in collaboration with the *Nature Computational Science* team. Peer reviewer reports are available.

## Answer ALS

**Emily G. Baxi**[7,8], **Alyssa N. Coyne**[7,8], **Elizabeth Mosmiller**[8], **Lindsey Hayes**[8], **Aianna Cerezo**[8], **Omar Ahmad**[8], **Promit Roy**[8], **Steven Zeiler**[8], **John W. Krakauer**[8], **Divya Ramamoorthy**[1], **Jonathan Li**[1], **Aneesh Donde**[1], **Nhan Huynh**[1], **Miriam Adam**[1], **Brook T. Wassie**[1], **Alex Lenail**[1], **Natasha Leanna Patel-Murray**[1], **Yogindra Raghav**[1], **Karen Sachs**[1], **Velina Kozareva**[1], **Stanislav Tsitkov**[1], **Tobias Ehrenberger**[1], **Julia A. Kaye**[9], **Leandro Lima**[9], **Stacia Wyman**[9], **Edward Vertudes**[9], **Naufa Amirani**[9], **Krishna Raja**[9], **Reuben Thomas**[9], **Ryan G. Lim**[10], **Ricardo Miramontes**[10], **Jie Wu**[11], **Vineet Vaibhav**[12], **Andrea Matlock**[12], **Vidya Venkatraman**[12], **Ronald Holewenski**[12], **Niveda Sundararaman**[12], **Rakhi Pandey**[12], **Danica-Mae Manalo**[12], **Aaron Frank**[13], **Loren Ornelas**[13], **Lindsey Panther**[13], **Emilda Gomez**[13], **Erick Galvez**[13], **Daniel Perez**[13], **Imara Meepe**[13], **Susan Lei**[13], **Louis Pinedo**[13], **Chunyan Liu**[13], **Ruby Moran**[13], **Dhruv Sareen**[13,14], **Barry Landin**[15], **Carla Agurto**[16], **Guillermo Cecchi**[16], **Raquel Norel**[16], **Sara Thrower**[5], **Sarah Luppino**[5], **Alanna Farrar**[5], **Lindsay Pothier**[5], **Hong Yu**[5], **Ervin Sinani**[5], **Prasha Vigneswaran**[5], **Alexander V. Sherman**[5], **S. Michelle Farr**[17], **Berhan Mandefro**[14], **Hannah Trost**[14], **Maria G. Banuelos**[14], **Veronica Garcia**[14], **Michael Workman**[14], **Richie Ho**[14], **Robert Baloh**[14], **Jennifer Roggenbuck**[18], **Matthew B. Harms**[19], **Carolyn Prina**[14], **Sarah Heintzman**[18], **Stephen Kolb**[18], **Jennifer Stocksdale**[20], **Keona Wang**[20], **Todd Morgan**[21], **Daragh Heitzman**[21], **Arish Jamil**[4], **Jennifer Jockel-Balsarotti**[22], **Elizabeth Karanja**[22], **Jesse Markway**[22], **Molly McCallum**[22], **Tim Miller**[22], **Ben Joslin**[23], **Deniz Alibazoglu**[23], **Senda Ajroud-Driss**[23], **Jay C. Beavers**[24], **Mary Bellard**[25], **Elizabeth Bruce**[25], **Jonathan D. Glass**[4], **Nicholas Maragakis**[8], **Merit E. Cudkowicz**[5], **James Berry**[5], **Terri Thompson**[26], **Ernest Fraenkel**[1], **Steven Finkbeiner**[9], **Leslie M. Thompson**[10,11,19,27], **Jennifer E. Van Eyk**[12], **Clive N. Svendsen**[13,14] and **Jeffrey D. Rothstein**[7,8]

[7]Brain Science Institute, Johns Hopkins University School of Medicine, Baltimore, MD, USA. [8]Department of Neurology, Johns Hopkins University School of Medicine, Baltimore, MD, USA. [9]Center for Systems and Therapeutics and the Taube/Koret Center for Neurodegenerative Disease, Gladstone Institutes and the Departments of Neurology and Physiology, University of California San Francisco, San Francisco, CA, USA. [10]UCI MIND, University of California Irvine, Irvine, CA, USA. [11]Department of Biological Chemistry, University of California Irvine, Irvine, CA, USA. [12]Advanced Clinical Biosystems Research Institute, The Barbra Streisand Heart Center, The Smidt Heart Institute, Cedars-Sinai Medical Center, Los Angeles, CA, USA. [13]Cedars-Sinai Biomanufacturing Center, Cedars-Sinai Medical Center, Los Angeles, CA, USA. [14]The Board of Governors Regenerative Medicine Institute, Cedars-Sinai Medical Center, Los Angeles, CA, USA. [15]Technome LLC, Herndon, VA, USA. [16]Computational Biology Center, IBM T.J. Watson Research Center, Yorktown, NY, USA. [17]Zofia Consulting, Reston, VA, USA. [18]Department of Neurology and Genetics, Ohio State University Wexner Medical Center, Columbus, OH, USA. [19]Department of Neurology, Columbia University, New York, NY, USA. [20]Department of Psychiatry and Human Behavior and Sue and Bill Gross Stem Cell Center, University of California Irvine, Irvine, CA, USA. [21]Texas Neurology, Dallas, TX, USA. [22]Department of Neurology, Washington University, St. Louis, MO, USA. [23]Department of Neurology, Northwestern University, Chicago, IL, USA. [24]Microsoft Research, Microsoft Corporation, Redmond, WA, USA. [25]Microsoft University Relations, Microsoft Corporation, Redmond, WA, USA. [26]On Point Scientific Inc., San Diego, CA, USA. [27]Department of Neurobiology and Behavior, University of California Irvine, Irvine, CA, USA.

**Pooled Resource Open-Access ALS Clinical Trials Consortium**

Alexander Sherman[5]

**ALS/MND Natural History Consortium**

**Christian Lunetta[28], David Walk[29], Ghazala Hayat[30], James Wymer[31], Kelly Gwathmey[32], Nicholas Olney[33], Senda Ajroud-Driss[34], Terry Heiman-Patterson[35], Ximena Arcila-Londono[36], Alexander Sherman[5], Kenneth Faulconer[5], Ervin Sanani[5], Alex Berger[5] and Julia Mirochnick[5]**

[28]Centro Clinico NeMO, Milan, Italy. [29]University of Minnesota, Minneapolis, MN, USA. [30]St. Louis University, St. Louis, MO, USA. [31]University of Florida—Gainesville, Gainesville, FL, USA. [32]Virginia Commonwealth University, Richmond, VA, USA. [33]Providence Brain and Spine Institute, Portland, OR, USA. [34]Northwestern University Feinberg School of Medicine, Chicago, IL, USA. [35]Temple University, Philadelphia, PA, USA. [36]Henry Ford Hospital, Detroit, MI, USA.

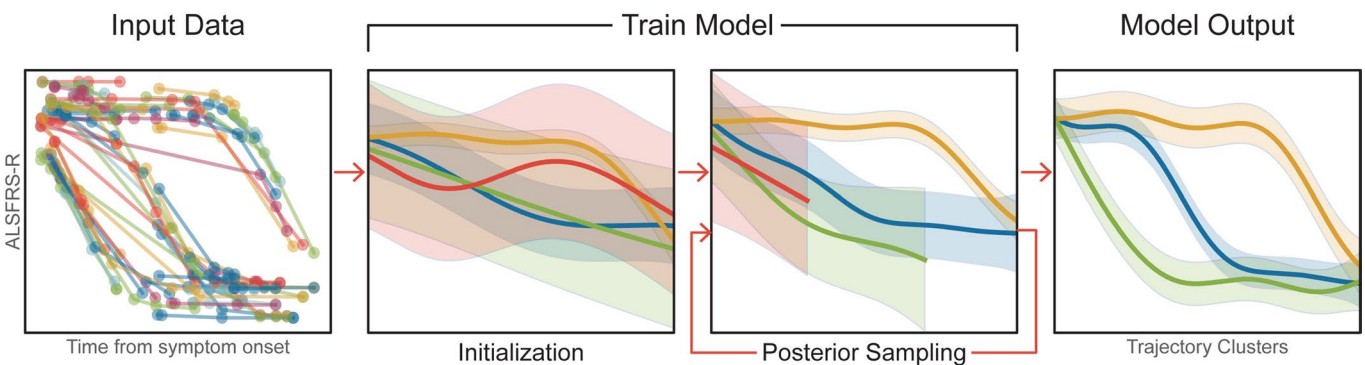

**Extended Data Fig. 1 | Model Workflow.** Input, training, and optimization of the Mixture of Gaussian Processes model.

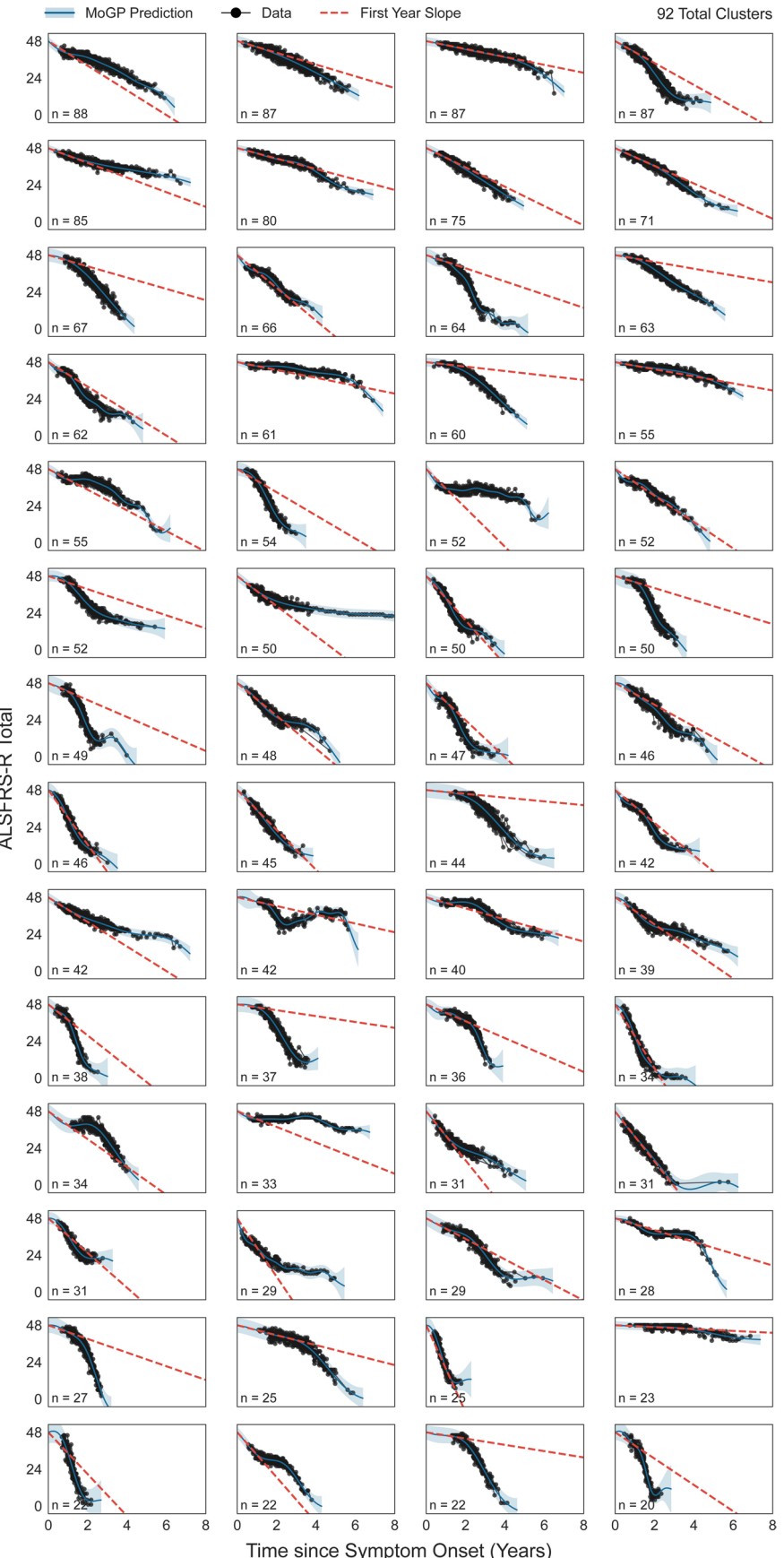

**Extended Data Fig. 2 | Clusters spanning 90% of all individuals in PROACT.** The first year slope is calculated as the difference between 48 and the mean cluster score one year after symptom onset, divided by the time from symptom onset. N indicates the number of ALS patients in each cluster. Shaded area indicates 0.95 confidence interval.

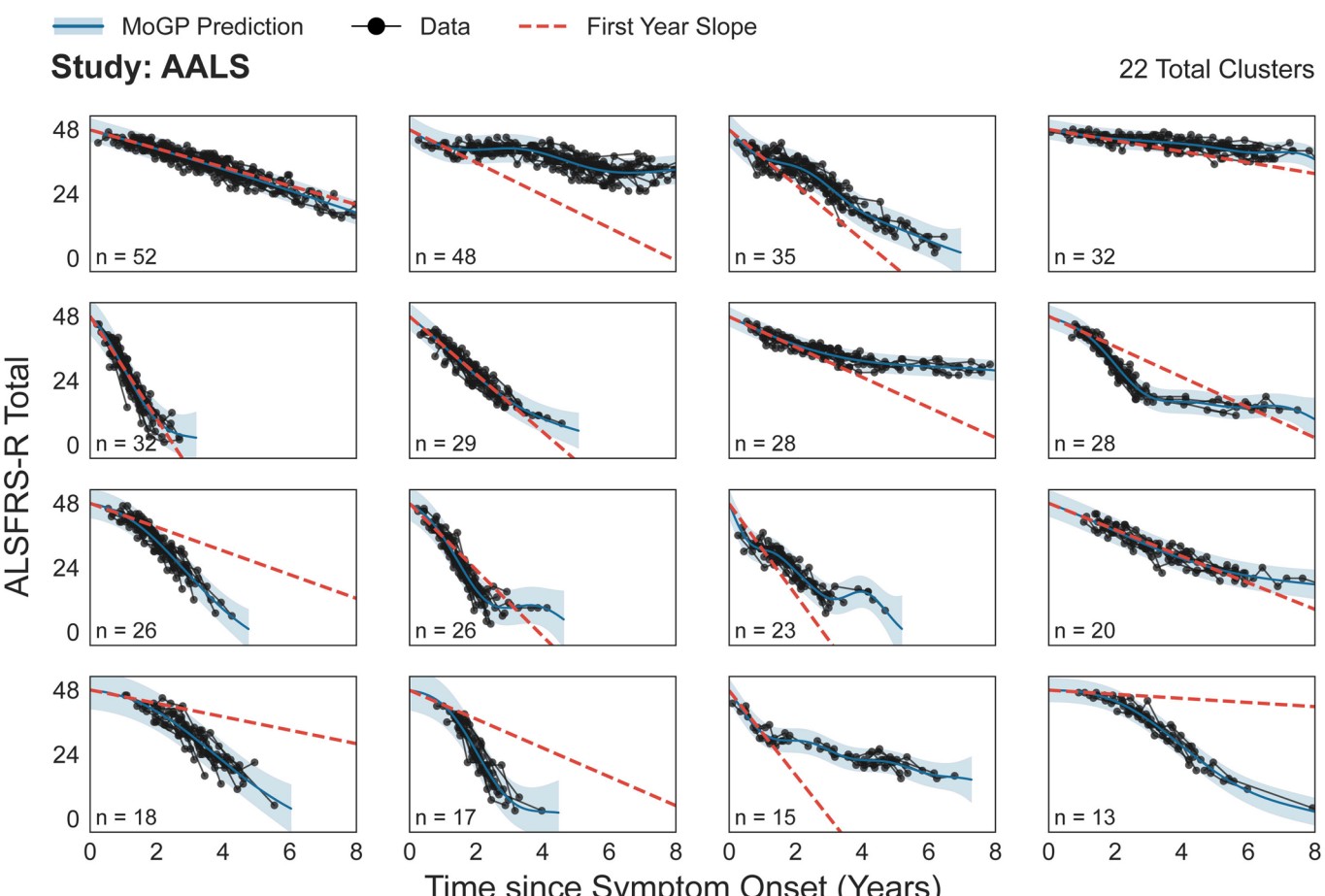

**Extended Data Fig. 3 | Clusters spanning 90% of all individuals in AALS.** The first year slope is calculated as the difference between 48 and the mean cluster score one year after symptom onset, divided by the time from symptom onset. N indicates the number of ALS patients in each cluster. Shaded area indicates 0.95 confidence interval.

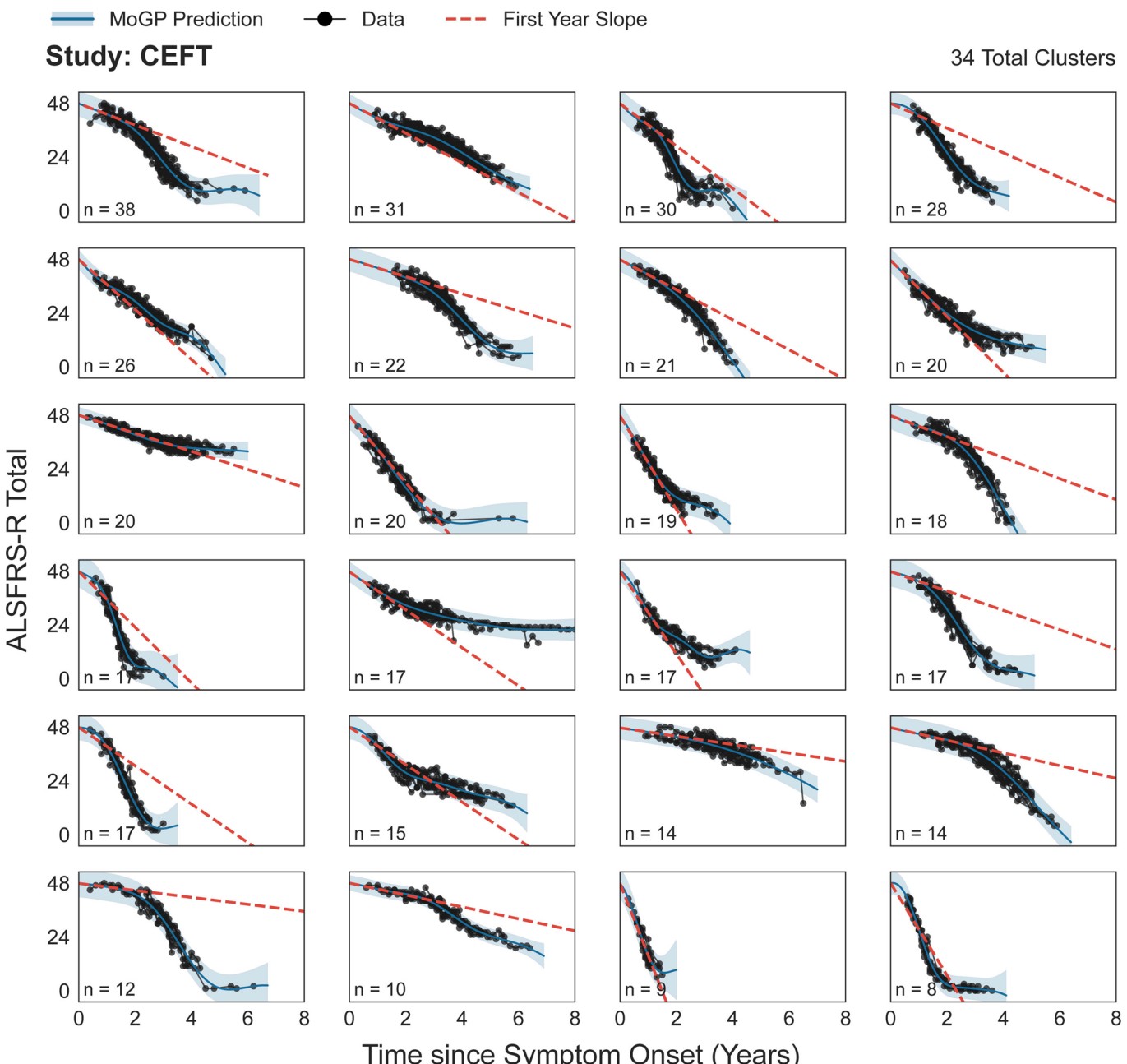

**Extended Data Fig. 4 | Clusters spanning 90% of all individuals in CEFT.** The first year slope is calculated as the difference between 48 and the mean cluster score one year after symptom onset, divided by the time from symptom onset. N indicates the number of ALS patients in each cluster. Shaded area indicates 0.95 confidence interval.

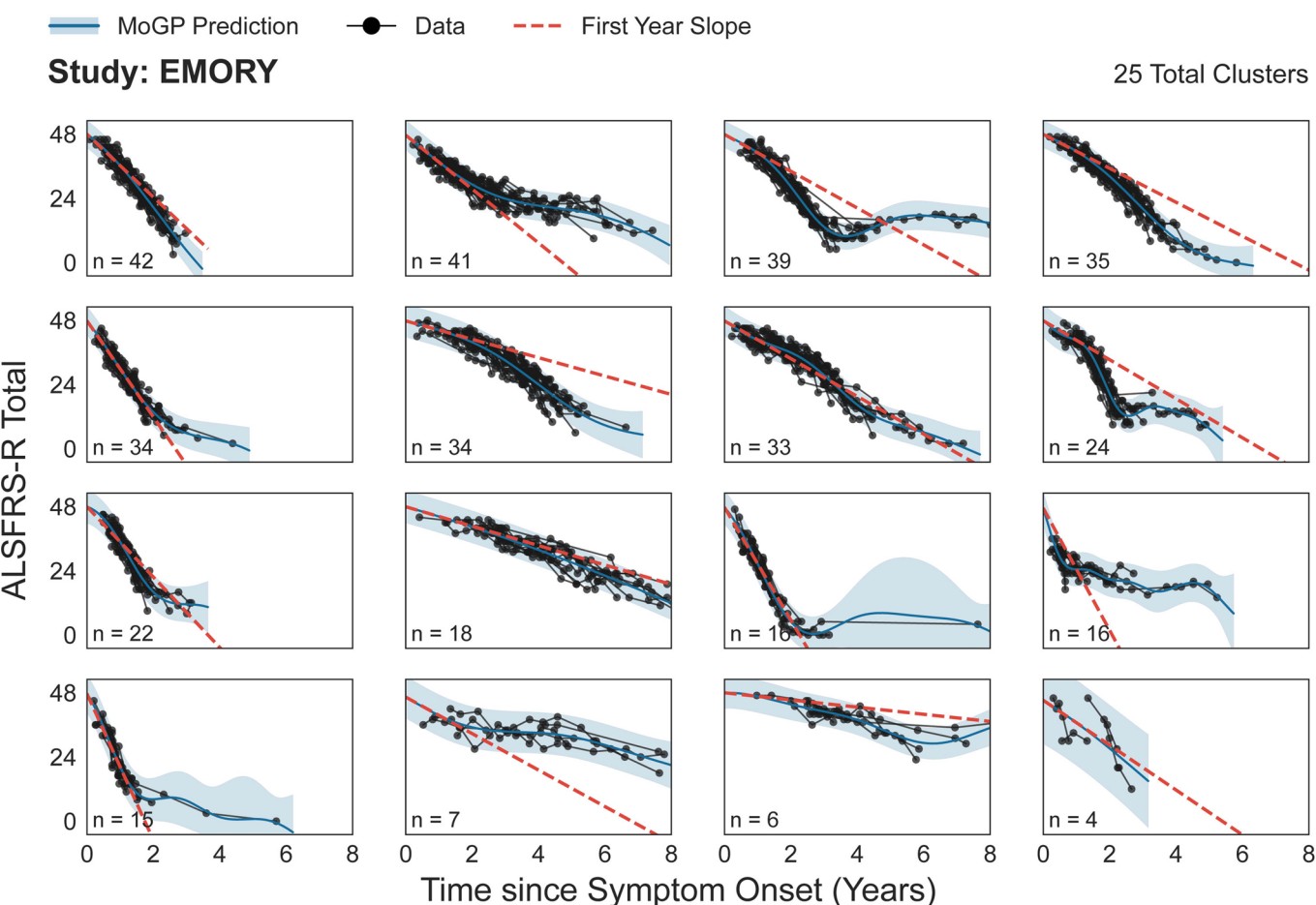

**Extended Data Fig. 5 | Clusters spanning 90% of all individuals in EMORY.** The first year slope is calculated as the difference between 48 and the mean cluster score one year after symptom onset, divided by the time from symptom onset. N indicates the number of ALS patients in each cluster. Shaded area indicates 0.95 confidence interval.

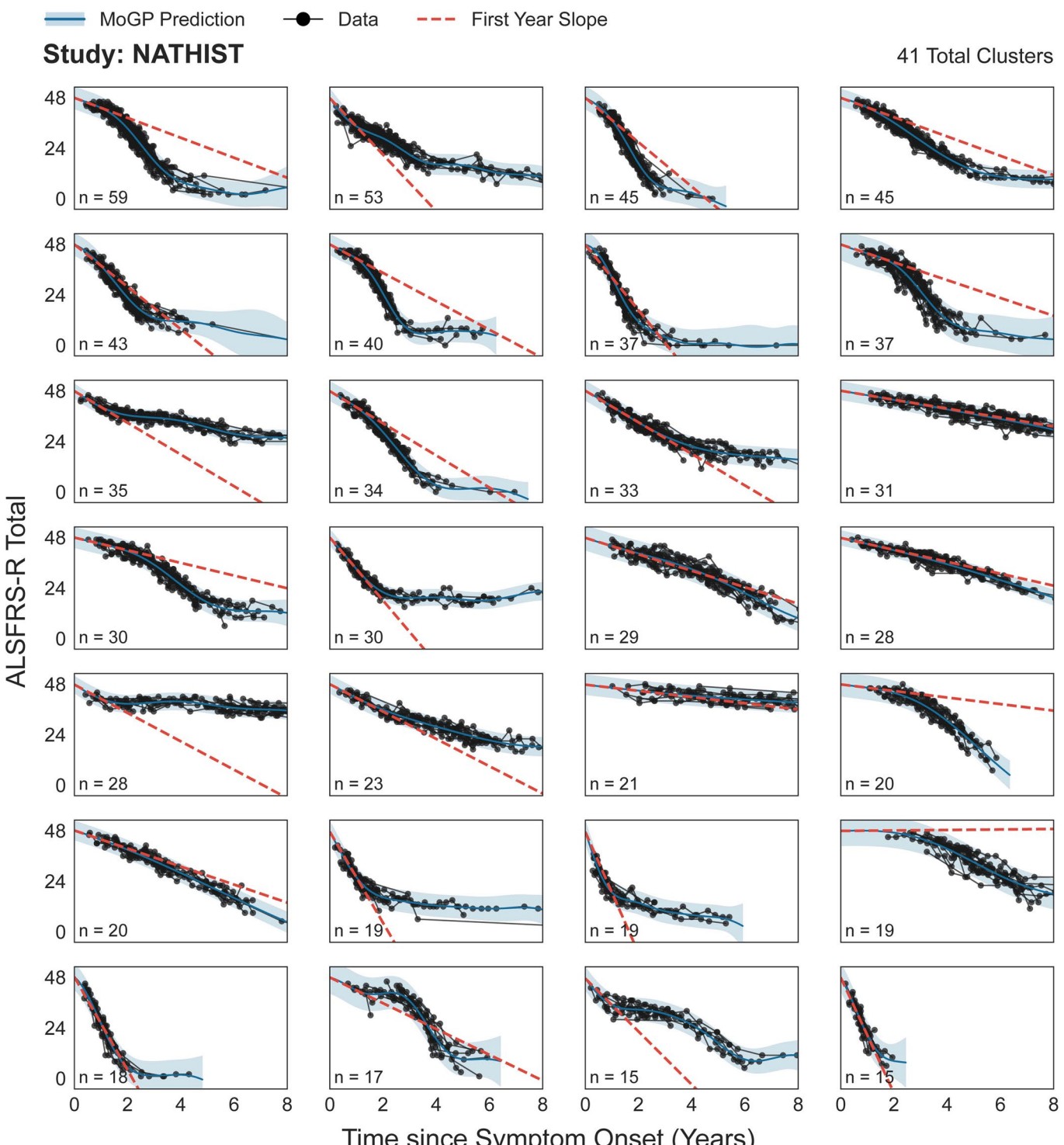

**Extended Data Fig. 6 | Clusters spanning 90% of all individuals in NATHIST.** The first year slope is calculated as the difference between 48 and the mean cluster score one year after symptom onset, divided by the time from symptom onset. N indicates the number of ALS patients in each cluster. Shaded area indicates 0.95 confidence interval.

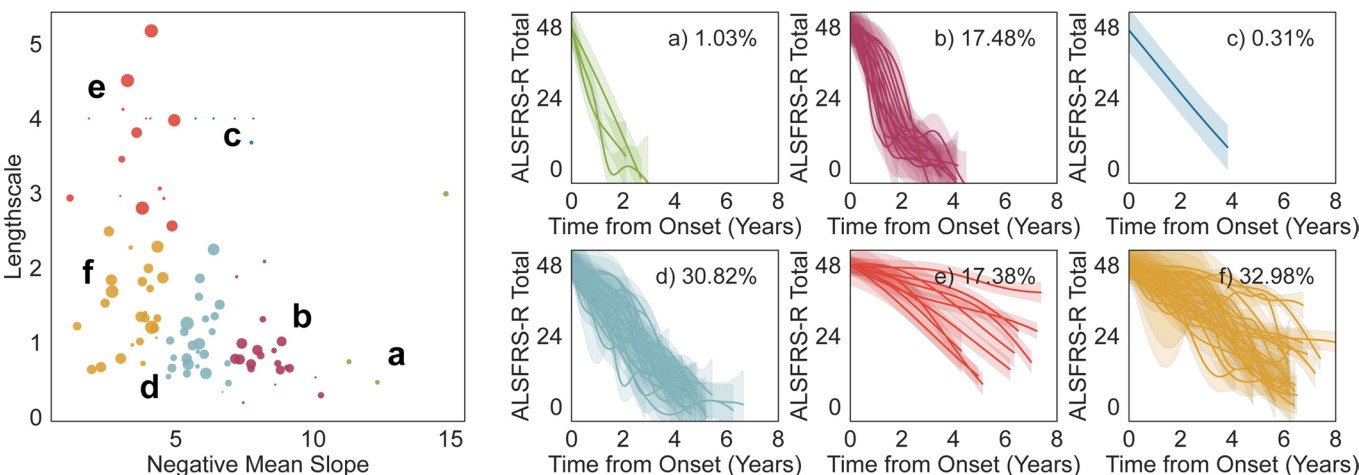

**Extended Data Fig. 7 | Dominant ALS progression patterns, identified using length-scale and negative mean function slope.** Length-scale indicates trajectory stability; negative mean function slope corresponds to rate of progression. Learned model parameters from the PRO-ACT reference model are k-means clustered (Left plot; k=6, marker size corresponds to cluster size), with clusters ≥ N=5 visualized, and percentage of individuals that fall within each of the trajectory patterns are labeled (Right plots).

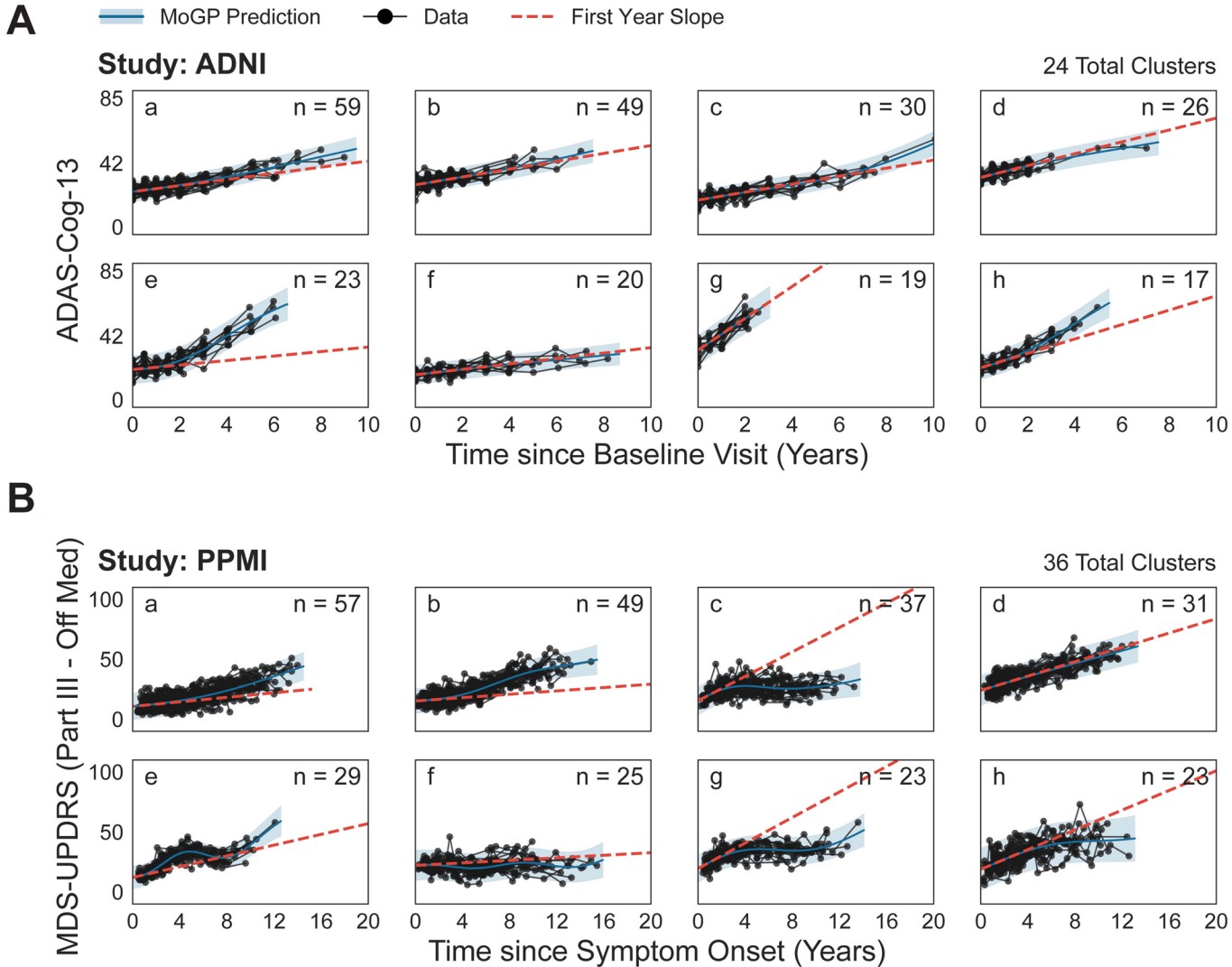

**Extended Data Fig. 8 | Identifying progression clusters from Alzheimer's and Parkinson's clinical measures.** Eight largest clusters are visualized. N indicates number of individuals in each cluster. The first year slope is calculated as: (mean cluster at one year – mean cluster score at initial value), divided by the time from the initial value.

# nature research

# Reporting Summary

Nature Research wishes to improve the reproducibility of the work that we publish. This form provides structure for consistency and transparency in reporting. For further information on Nature Research policies, see our Editorial Policies and the Editorial Policy Checklist.

## Statistics

For all statistical analyses, confirm that the following items are present in the figure legend, table legend, main text, or Methods section.

| n/a | Confirmed | |
|---|---|---|
| ☐ | ☒ | The exact sample size ($n$) for each experimental group/condition, given as a discrete number and unit of measurement |
| ☐ | ☒ | A statement on whether measurements were taken from distinct samples or whether the same sample was measured repeatedly |
| ☐ | ☒ | The statistical test(s) used AND whether they are one- or two-sided *Only common tests should be described solely by name; describe more complex techniques in the Methods section.* |
| ☐ | ☒ | A description of all covariates tested |
| ☐ | ☒ | A description of any assumptions or corrections, such as tests of normality and adjustment for multiple comparisons |
| ☐ | ☒ | A full description of the statistical parameters including central tendency (e.g. means) or other basic estimates (e.g. regression coefficient) AND variation (e.g. standard deviation) or associated estimates of uncertainty (e.g. confidence intervals) |
| ☐ | ☒ | For null hypothesis testing, the test statistic (e.g. $F$, $t$, $r$) with confidence intervals, effect sizes, degrees of freedom and $P$ value noted *Give P values as exact values whenever suitable.* |
| ☐ | ☒ | For Bayesian analysis, information on the choice of priors and Markov chain Monte Carlo settings |
| ☒ | ☐ | For hierarchical and complex designs, identification of the appropriate level for tests and full reporting of outcomes |
| ☒ | ☐ | Estimates of effect sizes (e.g. Cohen's $d$, Pearson's $r$), indicating how they were calculated |

*Our web collection on statistics for biologists contains articles on many of the points above.*

## Software and code

Policy information about availability of computer code

| Data collection | No software used to collect data. |
|---|---|
| Data analysis | We provide the python code for the MoGP framework as well as a pre-trained reference model that researchers can use to generate predictions of cluster membership and trajectory function from input patient data. We also provide a pip-installable Python package associated with this work (mogp). All code used for data processing, modeling, and figure generation can be found at: https://github.com/fraenkel-lab/mogp. Code also is deposited on Zenodo (License: BSD 3-Clause; https://doi.org/10.5281/zenodo.6744399).<br><br>Python version and specific package versions used for analysis listed below:<br>Python: 3.7.3<br>Packages:<br>joblib: 1.0.0<br>numpy: 1.19.4<br>pandas: 1.3.1<br>openpyxl: 3.0.5<br>sas7bdat: 2.2.3<br>seaborn: 0.11.1<br>statannot: 0.2.3<br>lifelines: 0.25.7<br>statsmodels: 0.12.2<br>mogp: 0.1.1<br>jupyter: 1.0.0<br>Gpy: 1.9.9<br>scipy: 1.7.3 |

April 2020

scikit-learn: 0.21.1
sklearn: 0.0
matplotlib: 3.1.1

Computational environments: Model run on Azure and compute cluster.
Azure specifications: Standard F32s_v2 machines (32 vCPUs, 64 Gb Mem)
Cluster: 16 cores, 1 node, 10GB memory

For manuscripts utilizing custom algorithms or software that are central to the research but not yet described in published literature, software must be made available to editors and reviewers. We strongly encourage code deposition in a community repository (e.g. GitHub). See the Nature Research guidelines for submitting code & software for further information.

## Data

Policy information about availability of data

All manuscripts must include a data availability statement. This statement should provide the following information, where applicable:
- Accession codes, unique identifiers, or web links for publicly available datasets
- A list of figures that have associated raw data
- A description of any restrictions on data availability

A pre-trained reference model for this study can be downloaded here: http://fraenkel.mit.edu/mogp

Source Data for Figures 2-4 and Extended Data Figure 7 are available with this manuscript. Source Data for Figure 1 and Extended Data Figure 2 are available as a Python object from http://fraenkel.mit.edu/mogp. Other source data are unavailable at this time due to containing patient-level clinical data; however, all figures can be generated using the code provided, after downloading the datasets listed below.

Clinical data for this study can be obtained from the following sources:
AALS (ClinicalTrials.gov Identifier: NCT02574390) is available for download in the AnswerALS data portal (data.answerals.org). PRO-ACT can be downloaded from the PRO-ACT database (https://nctu.partners.org/ProACT). CEFT (ClinicalTrials.gov Identifier: NCT00349622) can be downloaded from National Institute of Neurological Disorders and Stroke (NINDS) (https://www.ninds.nih.gov/Current-Research/Research-Funded-NINDS/Clinical-Research/Archived-Clinical-Research-Datasets). EMORY is restricted access at this time due to containing information that could compromise patient privacy, but available with permission from Dr. Jonathan Glass (jglas03@emory.edu) for legitimate research. Response to request will be provided within two weeks, all data provided will be fully de-identified, a DUA will need to be established, and the source data will need to be acknowledged in any publications. NATHIST is available from the ALS/MND Natural History Consortium (https://www.data4cures.org/requestingdata) with a summary of proposed data use, data elements requested, and publication intent. PPMI can be downloaded, with a data use agreement, online application, and compliance with publication policy (https://www.ppmi-info.org/access-data-specimens/download-data). Applications for data access are reviewed by the Data and Publications Committee within one week of receipt. ADNI can be downloaded through the LONI Image and Data Archive (https://adni.loni.usc.edu/data-samples/access-data/#access_data). Access is contingent on adherence to the ADNI Data Use Agreement and their publication policies. The application process includes the acceptance of a Data Use Agreement and submission of an online application form. The application must include the investigator's institutional affiliation and the proposed uses of the ADNI data. ADNI data may not be used for commercial products or redistributed in any way.

# Field-specific reporting

Please select the one below that is the best fit for your research. If you are not sure, read the appropriate sections before making your selection.

☒ Life sciences          ☐ Behavioural & social sciences          ☐ Ecological, evolutionary & environmental sciences

For a reference copy of the document with all sections, see nature.com/documents/nr-reporting-summary-flat.pdf

# Life sciences study design

All studies must disclose on these points even when the disclosure is negative.

| | |
|---|---|
| Sample size | Analysis was conducted using only de-identfied datasets from previously collected clinical cohorts. Data for this analysis was obtained from four large study populations. Three observational ALS studies were used: Answer ALS (N=456 patients; NCT02574390) the Emory ALS Clinic database (N=399 patients), the ALS/Natural History Consortia (N=907), and two overlapping clinical trial datasets: The Pooled Resource Open-Access ALS Clinical Trials (N=2923 patients) and the Clinical Trial of Ceftriaxone in ALS (N=476; NCT00349622). |
| Data exclusions | Data was preprocessed prior to analysis to select participants with consistent, longitudinal data available. Participants were excluded from the model if fewer than three complete ALSFSR-R visits were recorded, the first visit was more than seven years from symptom onset, or an increase of greater than six points in ALSFRS-R between subsequent visits was recorded. These criteria were pre-established. |
| Replication | We conducted extensive analysis to evaluate the robustness of our results. These included comparing model results across heterogeneous clinical cohorts, withholding data and evaluating the accuracy of reconstructing that data, and using test/train sets to evaluate model transferability between datasets. Our model demonstrates strong robustness across these settings, and these results are detailed in the manuscript. |
| Randomization | Because our analysis does not compare a treatment and control group, randomization is not relevant to this study. For controlling for covariates: by including multiple large clinical cohorts, we were able to evaluate model performance across common sources of covariates in clinical data, such as variations in rate of progression in each cohort, frequency of clinical visits, size of the clinical study, and clinical sites. For analysis using test/train datasets, the splits were randomly assigned. |

| Blinding | CEFT was a blinded study (neither participants nor study staff will knew which treatment a participant received). PRO-ACT is a large database that aggregates anonymized clinical trial data, without disclosing explicit blinding information. The observational studies used here were not blinded to any particular therapy. |
|---|---|

# Reporting for specific materials, systems and methods

We require information from authors about some types of materials, experimental systems and methods used in many studies. Here, indicate whether each material, system or method listed is relevant to your study. If you are not sure if a list item applies to your research, read the appropriate section before selecting a response.

## Materials & experimental systems

| n/a | Involved in the study |
|---|---|
| ☒ | ☐ Antibodies |
| ☒ | ☐ Eukaryotic cell lines |
| ☒ | ☐ Palaeontology and archaeology |
| ☒ | ☐ Animals and other organisms |
| ☒ | ☐ Human research participants |
| ☒ | ☐ Clinical data |
| ☒ | ☐ Dual use research of concern |

## Methods

| n/a | Involved in the study |
|---|---|
| ☒ | ☐ ChIP-seq |
| ☒ | ☐ Flow cytometry |
| ☒ | ☐ MRI-based neuroimaging |

