## [Peer Review File · Nature Computational Science]

Identifying Patterns of ALS Progression from Sparse Longitudinal DataDecision Letter, Reviewer comments, first version:

Subject: Decision on Guided Open Access manuscript GUIDEDO-21-00225

Message: 3rd January 2022

Dear Professor Fraenkel,

Thank you again for choosing to submit your manuscript using the Guided Open Access pilot at the Nature Portfolio. As part of this process, our editorial team has considered your paper for three of our journals with strong interest in publishing in your field: Nature Computational Science, Nature Communications, and Communications Biology.

Your manuscript entitled "Identifying Patterns of ALS Progression from Sparse Longitudinal Data" has now been reviewed by 3 experts in amyotrophic lateral sclerosis, whose comments are included below and in the attached Editorial Assessment Report. As part of the Guided Open Access pilot, editors from all 3 journals have discussed the reviewer reports and the manuscript's suitability for our journals. After careful evaluation, our editorial recommendation is to revise the manuscript and submit back through the Guided Open Access submission portal for consideration at Nature Computational Science, Nature Communications, or Communications Biology using the link provided below. Provided the revisions satisfy all technical and editorial concerns, all three journals are very interested in publishing your manuscript. Please see details in the attached Editorial Assessment Report.

In brief, for publication in Nature Computational Science, please address all comments put forth by the reviewers and also, provide additional experiments showing the applications of the proposed methodology to multiple domains to establish the broad applicability of the study.

Please note that the Editorial Assessment Report is a standalone document that contains an editorial evaluation, recommendation and portable peer advice to help you navigate and interpret the reviewers' reports. It also provides guidance for adhering to best practice with regard to transparency and reproducibility, for example on the issue of sharing data. We have also included information about data accessibility and reproducibility, which we hope you find useful.

HOW TO SUBMIT

Once you are ready to submit a revised version of your manuscript, please use the link below to submit the following items as separate documents:

- Revised manuscript. Please show all changes in the manuscript text file with tracked changes or colour highlighting
- Any supplementary files.
- Point-by-point response to the reviewers' comments, reproduced verbatim. If you are unable to address specific reviewer requests or find any points invalid, please explain why.
- Cover letter to the editor, stating the journal for which you have revised.

[REDACTED]

This URL links to your confidential home page and associated information about manuscripts you may have submitted, or that you are reviewing for us. If you wish to forward this email to co-authors, please delete the link to your homepage.

Should you have any questions about the recommended journals or would like advice on the revisions, you can contact me directly and I will be happy to assist. We look forward to receiving the revised version of your manuscript.

Yours sincerely,
Ananya Rastogi

--

Ananya Rastogi
Editor
Guided Open Access

On behalf of the Guided OA editorial team

Reviewer #1 (Remarks to the Author: Overall significance):

The authors present a mixture of Gaussian process (MoGP) method followed by Dirichlet modeling to predict ALS decline over time in sub-populations or "clusters" across 4 different cohorts. The overall method of time series prediction and clustering over longitudinal data, including temporal disease progression, is not novel and has been successfully performed in other diseases like Alzheimers (Peterson, et. al., NeurIPS, 2018), multiple sclerosis (Zhao, et. al., 2015, IEEE Conference in Data Mining), and longitudinal omics (Cheng, et. al., 2019, Nature Communications), and many others. While the methods here are not novel, their application does further solidify the hypothesized non-linearities present in clinical ALS. Presently, the authors have over-represented the clinical significance of their clusters given that, unlike other biomedical MoGP models, the authors were not able to provide clear interpretability of the clusters. The developed model is of interest to the ALS field, although revisions are suggested to better frame the method and results in a manner that realistically portrays current significance.

1. While MoGP has not been the focus of prior ALS models, it has been used in other similar temporal disease predictions, including identifying sub-populations based on disease progression. The authors should briefly mention similar use cases in other diseases. This is important context, particularly for readers who may not have a machine learning background.
2. The results consist of 3 main aspects: prediction of ALS decline using MoGP, identifying of clusters of patients with similar progression patterns, assessing non-linearity. While these tasks are inter-related from a method standpoint, the results need to be better separated to reflect the different objectives from an ALS domain standpoint. This could be done with structural format and headings, as well as order of presentation. First discuss the ability of the model to predict a given ALS patient's progression. Then discuss the clustering. Finally, discuss the presence of linear and non-linear clusters. A sub-section of the last section would be comparing the MoGP results to the linear slope models and other linear methods of ALS prediction previously utilized in the literature.
3. Currently the authors are comparing their population MoGP model results to the linear models for individual patients (patient slope models). This makes sense for a sub-section emphasizing the importance of having a method like MoGP that is "flexible" and can model both the predominantly non-linear progressions as well as the smaller portion of linear trajectories. However, it was surprising that the authors did not include the most obvious baseline: making a personalized gaussian regression model for each patient and then assess the population-level MoGP generalizability as compared to the Gaussian models for each individual patient. This would be a more apple-to-apples comparison for the sake of generalizability.
4. The clustering of progression patterns is certainly of clinical interest and significance. However, the clinical significance of the cluster results are over-stated. The authors do not make clear domain connections to the large numbers of clusters. The only domain content indicated by clusters was the "cliff", linear, and sigmoid hypotheses. Supplementary Table 5 indicates there is a significantly different number of clusters as a function of sample size. If more clear connections to domain features cannot be made within the scope of the present work, the authors need to simply pull back on their language and note that connecting features to the clusters would be part of future work.
5. The authors do not directly address why the previous models are more accurate than the present MoGP with less training data years. This reviewer suspects it has to do with the mixing parameter. This could be easily evaluated with a parameter sensitivity analysis. Once proven, this result would add additional credibility to the MoGP model presented and help provide better constraints as to what is needed (sample size, training years, visits per patient, etc.) to make the MoGP model best suited for future predictions compared to prior ALS models.
6. Most machine learning/AI modeling papers have a model workflow or pipeline figure that clearly articulates the steps of the workflow and/or involved algorithm(s). Such a figure would really help this work. Additionally, have a pseudocode table or figure with more pertinent algorithm details in the supplement would help...particularly for training, parameter tuning, and optimization steps.
7. MINOR: The authors need to revisit the technical language. The use of first person language and pronouns throughout is more in line with an IEEE conference proceeding than a clinical or domain journal.

Reviewer #1 (Remarks to the Author: Impact):

The presented MoGP model definitely adds to present discussion in the field that ALS progression is predominantly non-linear. The overwhelming number of non-linear clusters is the most impactful result. However, the paper in its present form, does not compare to enough baselines to illustrate outright ALS progression prediction superiority across the board - in other words, it does not prove it's the best ALS model out there. The clusters are also very interesting in terms of clearly illustrating the preponderance of non-linear progressions, but they fail to fully connect to domain features that a clinical audience will appreciate. The authors do write some text with a couple of cluster examples and how they map to survival; however, more work is needed to make this a key point [if the authors want this to be a key point within their present scope of work]. In summary, focusing more on the non-linear result (which is clearly and quantitatively proven) is the strongest part of the work.

However, that result is somewhat buried in the present text. Restructuring would help emphasize this finding more and minimize some of the less impactful areas where future work is still needed.

Reviewer #1 (Remarks to the Author: Strength of the claims):

1. The authors should compare the population MoGP model to single patient Gaussian regression models. This is more of an apples-to-apples comparison. The comparison to the linear slope models and such should only be used to emphasize the necessity to model non-linearity.
2. If the authors want to make the clusters be central to their work beyond illustrating the number of clusters that were non-linear progression versus linear, more detail and context needs to be given to the clusters' ties to clinical metrics. The few sentences with sparse examples on survival and respiratory function are not enough. OR the authors need to tone the language down on the significance of the clusters to only focus on importance of non-linearity and then write a limitations and future directions section to discuss future mapping of clusters to other clinical variables/features, citing basic examples there.

Reviewer #1 (Remarks to the Author: Reproducibility):

1. The authors need to provide more information on training and optimization protocols. Pseudocode tables would be helpful context.
2. While the code will be provided, some more details are necessary in the paper. Also, the authors give no detail in the paper on the types of software packages used, what type of computational environment the model was run on, etc.

Reviewer #2 (Remarks to the Author: Overall significance):

In this study, the authors proposed a new approach to quantify disease progression in ALS. Since linear models are not ideal, the authors explored aggregating patient trajectories in individualized clusters, each with a specific course, regarding rate and curve features.

Overall, this text is not simple to be followed by most neurologists caring ALS patients. To reach greater clinical impact some technical simplification is recommended, if this is the target.

The authors used 4 databases. Three are relatively small, Answer ALS, CEFT and EMORY, regarding the first we are not aware who introduced the data in the site (patients?), concerning the latter, the very fast rate of decline indicates that it represents a quite specific group of patients. Two databases (PRO-ACT and CEFT), they partially overlap, which is probably not a good solution regarding training and validation of their model. Other large databases are available, in particular in Europe (Westeneng HJ, et al, Prediction of personalised prognosis in patients with amyotrophic lateral sclerosis: development and validation of a prediction model, Lancet Neurology 2018), which could be used in this study.

For this model the authors do not mention the number of required patients for its development. Exclusion criteria are very loose and arbitrary. Patients with a first visit more than 24 or 36 months after disease onset would not be accepted in a trial (the authors propose that their tool could be used in future clinical trials), they decided for 7 years; and improvement of ALSFRS-R greater 6 points is never observed in an ALS clinic (if the diagnosis is correct), why not 2 or 3, considering and acceptable fluctuation? Did they include patients with PEG or NIV at entry?

Results are good. Regarding survival, it would be convenient to compare outcome of their model with the ones published applying different models (Westeneng HJ, et al, Prediction of personalised prognosis in patients with amyotrophic lateral sclerosis: development and validation of a prediction model, Lancet Neurology 2018). Some results using FVC predicted value and ALSFRS-R subscore were mentioned in results (see figures), but there no relevant information in methods about these analyses.

Discussion is appreciated.

Minor

Last paragraph of the Introduction summarized the article, which is not necessary.

The text is somewhat repetitive in some parts, for example last paragraph on page 4 is replicated in Modelling Approach on the next page, and PRO-ACT features are described on pages 9 and 10.

Reviewer #2 (Remarks to the Author: Impact):

This is a good work, with potential great impact. Possibly Nature Computational Science would be the best room.

Reviewer #2 (Remarks to the Author: Strength of the claims):

ALS is a very competitive area, and computational modelling is a new exciting field. After revision, this manuscript has a great chance of a relevant impact. To use another large data base in addition to PRO-ACT would strengthen their conclusions, they used 3 other relatively small, and one with overlapping with PRO-ACT.

Reviewer #2 (Remarks to the Author: Reproducibility):

I believe this could be reproduced by other authors.

Reviewer #3 (Remarks to the Author: Overall significance):

This study provides a characterisation of the longitudinal trajectory of the ALSFRS-R in amyotrophic lateral sclerosis. The developed model was also validated in other datasets. The result is original and can be applied to other fields where longitudinal data is available and behaves non-linear.

Reviewer #3 (Remarks to the Author: Impact):

Because of the complexity of the model (which I personally appreciate and that is explained and investigated well by the authors) I have some doubts about the implementation in practice.

Reviewer #3 (Remarks to the Author: Strength of the claims):

Ramamoorthy et al. studied the longitudinal trajectory of the revised version of the amyotrophic lateral sclerosis functional rating scale (ALSFRS-R). The authors developed and validated a sophisticated Bayesian non-linear model for the longitudinal trajectory of the ALSFRS-R. Reviewing this well-performed study was a great pleasure but I have also some comments aiming to further improve this study.

1. The authors frame their study as 'modelling ALS progression'. ALS progression is, however, much broader than patients daily functioning which is measured by the ALSFRS-R. It would be great if the authors could be clearer about this throughout the abstract and manuscript.
2. I agree with the authors that characterizing heterogeneity in ALS is important but the last sentence of the abstract 'Our results provide a critical advance in characterizing the heterogeneity in disease progression patterns of' is somewhat overstated. This also applies to the last sentence of the introduction.
3. Minor. In paragraph 2 of the introduction, the authors discuss the change in ALSFRS-R slope that is used in clinical trials. They classify ~ 0.4 points difference as a small effect, but given that the average decline of the ALSFRS-R is 0.5-1.0 points per month in population-based datasets (in trial populations it might be somewhat higher) this needs to be adjusted. Moreover, edaravone is not approved in Europe.
4. Minor. In the first sentence of the fourth paragraph of the introduction a typo might have occurred ('the more a more').
5. Table 1. The distribution of 'number of visits' and 'months followed' can be very skewed. A median and range (or interquartile range) would be more appropriate. It is unclear how the ALSFRS-R slope was calculated in this table. This is important because the degree of decline can be very skewed and, if possible, a more robust measure of this slope would be preferred over the mean and standard deviation. Finally, the number of characteristics provided is too little to be sufficiently informed about the datasets used. In summary, more detail is needed.
6. Around 3000 participants from the PRO-ACT database were included (which was by far the largest dataset used). Can the authors please comment on this selection? Which criteria were used to select these patients from the PRO-ACT database and why? What happened when more patients from the PRO-ACT database were included?
7. Subjects with at least 4 visits were used for prediction and subjects with at least 10 visits were used for assessing interpolation. Could the authors please provide analyses of what happened when fewer data points were available (to really demonstrate how robust each model is to sparse data)? This is even more important because (even from a trial population with usually a lot of measurements, i.e. PRO-ACT) >50% of the subjects were excluded because they have less than the data points needed. And could the authors please evaluate not only the MoGP, SM and LKM model but also the sigmoidal model? And which results were obtained when performing these analyses in the other datasets?
8. When reading the methods section about 'Model Generalizability' I first interpreted that the model developed in the PRO-ACT database (i.e. reference model) was modified before it was applied to other datasets. After reading the results it became clear that this was not the case. Could the authors please clarify that the primary analysis was to develop a model in the PRO-ACT database and apply this unchanged to the other datasets? The 'study specific models' can be mentioned as additional sensitivity analyses to investigate possible overfitting of the reference model. And as a minor comment, it could be added that the test and train datasets were split randomly.

9. From the methods it was somewhat unclear how the 'monotonic inductive bias' was incorporated. After reading the supplement this became clear for the 'negative linear mean function', but I still have some difficulties with interpreting the 'threshold function'. Could the authors please consider improving the section about 'monotonic inductive bias'?

10. The authors claim: 'The heterogeneity of the populations enabled us to measure the robustness of our model to data collection methods and the generalizability of ALS progression patterns between varying study populations'. This claim about heterogeneity is very much dependent on the underlying causes of this heterogeneity and not so much on the few measures provided (ALSFRS-R slope and follow-up duration). This refers also to my comment about table 1.

11. In supplementary figure 1 different clusters of the non-PRO-ACT dataset are plotted. These figures, however, display only a relatively small part of the data: AALS 284 of 456 patients (62%), CEFT 216 of 476 (45%) and EMORY 282 of 399 (71%). Could the authors please show an increased number of figures to demonstrate the different clusters? I would suggest that clusters of at least 90% of the data would be provided. This also applies to the clusters found in the PRO-ACT database (figure 1) which includes 1573 out of 2923 patients (54%). If Figure 1 becomes too large it possibly can be provided supplementary. Moreover, different clusters look very similar. Can the authors please provide a similarity score between clusters?

12. 92 clusters were found using the PRO-ACT data, while in the other datasets a maximum of 34 clusters was found. This is intrinsically related to the methods used but the meaning of this difference needs to be discussed in the discussion.

13. Minor. Based on the text, the 'greater than sign' in supplementary tables 2, 3 and 4 should be replaced by a 'greater than or equal to sign'.

14. Minor. I think that figure 2a and 2b have different messages and it might result in interpretation difficulties to combine them. Maybe these figures can be split up into separate figures?

15. Figure 3. These figures are now somewhat difficult to read, especially the error bars (which are very small). I think that it would be much more straightforward to interpret when the authors plot the full distribution of the (absolute) deviation of points from the modelled mean (if needed with a log or square root transformation). Some readers might for example interpret the first blue bar in Figure 3A as an error of 3 points (with very small error bars) which could be interpreted by some readers as that the model has nearly always an error of 3 points, which is a lot. Significant differences with other models have only a very limited meaning because all these models perform suboptimally. This interpretation can be prevented by just plotting the absolute differences and it provides also more insight into the full distributions of errors.

16. Minor. In the supplement, the authors describe the α parameter, which indicates the scaling parameter of the beta prior, but no value for this parameter is provided.

17. I could not find a discussion of possible limitations. Besides my suggestions above, I think that the lack of population-based datasets can be seen as a potential limitation and could lead to selection bias. Furthermore, attrition bias is a common problem in ALS research. Could the authors please discuss these two biases that could be present in their study and what the meaning of these biases is for the interpretation of their study?

Reviewer #3 (Remarks to the Author: Reproducibility):

The analyses were appropriate. The developed model was validated in other datasets. The code for this study is provided online.

Author rebuttal, first version:

Reviewer #1 (Remarks to the Author: Overall significance):

The authors present a mixture of Gaussian process (MoGP) method followed by Dirichlet modeling to predict ALS decline over time in sub-populations or "clusters" across 4 different cohorts. The overall method of time series prediction and clustering over longitudinal data, including temporal disease progression, is not novel and has been successfully performed in other diseases like Alzheimers (Peterson, et. al., NeurIPS, 2018), multiple sclerosis (Zhao, et. al., 2015, IEEE Conference in Data Mining), and longitudinal omics (Cheng, et. al., 2019, Nature Communications), and many others. While the methods here are not novel, their application does further solidify the hypothesized non-linearities present in clinical ALS. Presently, the authors have over-represented the clinical significance of their clusters given that, unlike other biomedical MoGP models, the authors were not able to provide clear interpretability of the clusters. The developed model is of interest to the ALS field, although revisions are suggested to better frame the method and results in a manner that realistically portrays current significance.

We thank the reviewer for their feedback and have more fully cited the prior work (see details below). In the revised manuscript, we also provide a higher-level clustering of the trajectories that does provide clear interpretability (See Rev 3, #11; Supp Fig. 7).

1. While MoGP has not been the focus of prior ALS models, it has been used in other similar temporal disease predictions, including identifying sub-populations based on disease progression. The authors should briefly mention similar use cases in other diseases. This is important context, particularly for readers who may not have a machine learning background.

We have added the discussion (below) of these and related modeling approaches in the methods section:

"The modeling approach of clustering over temporal progression patterns has been shown to improve characterization of disease progression in other conditions. For example, Peterson et al. demonstrated the use of an auto-regressive Gaussian Process model for predicting metrics of Alzheimer's progression; however the model makes a fundamentally different assumption about the structure of the data – that there is a single global progression type, and that each patient follows a noisy version of this global progression type – which is an assumption that does not capture the full heterogeneity of ALS phenotypes.³³ Furthermore, the model requires fixed time intervals of visits, which are not available in many clinical ALS datasets.³³ Zhao et al. present a related clustering approach in Multiple Sclerosis, although their model relies heavily on prior domain knowledge on how to group patients into subgroups, which has not as yet been clearly defined in ALS.³⁴ Other related models, like additive Gaussian process regression³⁵, can be used to characterize patterns in time series data, although they lack the ability to stratify patients into disease subtypes."

2. The results consist of 3 main aspects: prediction of ALS decline using MoGP, identifying of clusters of patients with similar progression patterns, assessing non-linearity. While these tasks are inter-related from a method standpoint, the results need to be better separated to reflect the different objectives from an ALS domain standpoint. This could be done with structural format and headings, as well as order of presentation. First discuss the ability of the model to predict a given ALS patient's progression. Then discuss the clustering. Finally, discuss the presence of linear and non-linear clusters. A sub-section of the last section would be comparing the MoGP results to the linear slope models and other linear methods of ALS prediction previously utilized in the literature.

We appreciate the reviewer's suggestions about organization, and we have adopted most of the reviewer's recommended structure, as follows: 1) Discussing the clustering 2) Discussing the presence of linear and non-linear clusters 3) Showing robustness of those clusters 4) Showing varied applied use-cases of the model.

We chose not to begin with patient-specific prediction, as the reviewer suggested. As we wrote in the original manuscript, if a user specifically wants a patient-specific prediction tool for prognosis, we recommend existing models directly suited for this purpose. This task has already been the subject of two large DREAM challenges in ALS (Kueffner 2019, Kuffner 2014). These models often focus on making predictions based on a number of features that are present at or close to diagnosis.

By definition, since we require longitudinal ALSFRS-R scores, we are creating a model that is not intended to be applied at diagnosis. The prediction of ALS decline is not presented as a major result, but instead intended to be a confirmation of cluster robustness to sparse data. Despite this not being an explicitly learned outcome of our model, we still see prediction as an important experiment to conduct - if the model were unable to predict future data, that would be an indicator that our clusters

were overfit to training data. This choice was the motivation for characterizing our prediction experiments under the heading “Disease progression trajectory clusters are robust to sparse data”.

3. Currently the authors are comparing their population MoGP model results to the linear models for individual patients (patient slope models). This makes sense for a sub-section emphasizing the importance of having a method like MoGP that is "flexible" and can model both the predominantly non-linear progressions as well as the smaller portion of linear trajectories. However, it was surprising that the authors did not include the most obvious baseline: making a personalized gaussian regression model for each patient and then assess the population-level MoGP generalizability as compared to the Gaussian models for each individual patient. This would be a more apple-to-apples comparison for the sake of generalizability.

We have added the personalized GP model to our supplement (Supp Fig. 4).

As a note, the evaluation method we use is biased towards per-patient models. For the purposes of evaluation, we use the root mean squared error between withheld data-points and the predicted function for that patient. In the case of a per-patient model, that function is only fit on data from that patient; in the case of the clustered models, the mean of the cluster (which sometimes includes up to 80 patients) is used. We expect a model fit on an individual patient to have a smaller RMSE than a model fit on a large cluster IF the modeling assumptions of the per-patient model match the data structure. One benefit of our model is that we can discover these modeling assumptions.

When fitting trajectories, the model must balance both being descriptive enough to capture data structure while not overfitting to noise in the data. A personalized GP tends to have similar or worse performance than a clustered GP model, given that a personalized GP will quickly revert to its mean function if no data is provided (included as Supp. Fig 4., copied below).

Beyond the previously included D50 sigmoidal model, we add the following additional patient-specific models: a quadratic model, and a linear mixed model. Of the tested parametric models, if a user wants a patient-specific model, we recommend the D50 sigmoidal model, with the following caveat previously noted in the results: “Since there are many settings in which patient-specific parametric models are very useful, we compared our model with a patient-specific sigmoidal model (SG).²⁰ Somewhat surprisingly, despite the fact that the MoGP models groups of patients, rather than individuals, MoGP outperforms a patient-specific sigmoid model by one or more ALSFRS-R points for 4.20%-9.43% of patients across the studies (Supplementary Table 4). This indicates that while a sigmoidal model captures much of the non-linearity, it does not represent the full complexity of progression patterns.”

PRO-ACT: Interpolation

PRO-ACT: Prediction

CEFT: Interpolation

CEFT: Prediction

Supplement Figure 4. Comparing model performance for interpolation and prediction against additional patient-specific baseline models. *p-val ≤ 1e-1 **p-val ≤ 1e-2 ***p-val ≤ 1e-3 ****p-val ≤ 1e-4 (Wilcoxon signed-rank one-sided test). Error bars show 0.95 confidence interval.

Further implementation details about these baseline models can be found in the supplement (copied below):

“Baseline Models:

Since there are many settings in which patient-specific parametric models are very useful, we provide additional characterization of per-patient parametric models using our framework: a personalized-GP, the D50 sigmoidal model^{20,21}, a quadratic model, and a linear mixed effect model.

Our personalized-GP model is initialized with the same priors as our MoGP model, and optimized using GPy (see Methods Supplement Table 1 for GP model priors).

Our D50 model is implemented in the following form:

$$y = \frac{48}{1 + e^{\frac{(x-D50)}{dx}}}$$

where D50 = time point when ALSFRS-R drops to 24; dx = slope of ALSFRS-R decrease. The model parameters are fit using `scipy curve_fit` (dogbox method, with bounds ((0.1, 0.1), (75, 5)), with a D50 initial value of 5, and a dx initial value of 0.5.

Our quadratic model is implemented in the following form:

$$y = ax^2 + bx + c$$

where coefficients a, b, and c are fit using `scipy curve_fit` (dogbox method, initial values a=1, b=1, c=1, no bounds).

Our linear mixed model is implemented using `statsmodels.formula.api.mixedlm`, with the design “Y~X”, and groups indicating individual patients. The model is fit using the `lbfgs` method.”

4. The clustering of progression patterns is certainly of clinical interest and significance. However, the clinical significance of the cluster results are over-stated. The authors do not make clear domain connections to the large numbers of clusters. The only domain content indicated by clusters was the "cliff", linear, and sigmoid hypotheses. Supplementary Table 5 indicates there is a significantly different number of clusters as a function of sample size. If more clear connections to domain features cannot be made within the scope of the present work, the authors need to simply pull back on their language and note that connecting features to the clusters would be part of future work.

We appreciate the reviewers concerns and have addressed them in two main ways. First, we do tone-down some of the claims in the manuscript. Second, we provide an approach to simplify the number of clusters to aid interpretability. Below are more details on both these approaches.

We have pulled back on language and add a limitations and future directions section:

“Although we have briefly shown correlations between the clusters and clinical features here, future directions can involve connecting the identified clusters to additional metrics, such as upper motor neuron function and NIV status. Going beyond our example of correlation of c9orf72 with clusters, further work can also involve connecting these to molecular measures of disease progression; this effort could yield molecular biomarkers for ALS, which currently are limited.”

We also provide the following to more clearly illustrate the model’s connections to domain features: 1) We provide additional domain classification of the trajectories, focusing on providing model metrics that allow a clinical audience to select clusters based on desired clinical properties of stability of the progression as well as the overall rate of progression [see below]. 2) We add two example applications in an Alzheimer’s and Parkinson’s dataset, to illustrate uses of the model clusters in non-ALS domains [see below; reviewer request].

Regarding the reviewer’s note that “there are a significantly different number of clusters as a function of sample size”, we now clarify in the text that the different number of clusters as a function of sample size is an expected property of Dirichlet processes: “One of the properties of the Dirichlet process model underlying MoGP clustering is that it will naturally scale the number of identified clusters within a given dataset depending on the number of samples in that dataset; in particular, one can use a reference model on a clinical cohort of any size. This non-parametric property of the model underlies the difference in total number of clusters found in the varying datasets.”

However, we recognize that it is often useful to analyze data in a small, fixed number of groups. We demonstrate that simply applying k-means clustering to the learned mean slope and lengthscale of the kernel function provides clearly interpretable dominant progression patterns which now appear in Supplemental Figure 7. The following analysis has been added to the text: “Clustering trajectories based on the optimized slope and lengthscale parameters reveals interesting patterns (Supp. Fig. 7). This analysis highlights the dominant progression patterns in ALS: sigmoidal fast progression (Supp. Fig 7B, 17.48% of individuals), stable slow progression (Supp. Fig 7E, 17.38%), unstable slow progression (Supp. Fig 7F, 32.98%), and unstable medium progression (Supp. Fig 7D, 30.82%). Clusters following a sigmoidal fast progression pattern have the highest percentage of individuals with bulbar onset (30.14% of individuals), while the stable slow progression pattern has the highest percentage of individuals with limb onset (76.97%) (Supp. Table 5). The median ages of onset for the patterns ranges between 54 to 59 years (Supp. Table 5).”

Supplemental Figure 7. Dominant ALS progression patterns, identified using lengthscale and negative mean function slope. Lengthscale indicates trajectory stability; negative mean function slope corresponds to rate of progression. Learned model parameters from the PRO-ACT reference model are k-means clustered (Left plot; $k=6$, marker size corresponds to cluster size), with clusters $\geq N=5$ visualized, and percentage of individuals that fall within each of the trajectory patterns labeled (Right plots).

The future direction of connecting molecular features to the clinical clusters is of particular interest to us, and is the basis for a large ongoing project aiming to identify the extent to which blood-based metabolomic biomarkers correlate with patterns of clinical trajectories.

5. The authors do not directly address why the previous models are more accurate than the present MoGP with less training data years. This reviewer suspects it has to do with the mixing parameter. This could be easily evaluated with a parameter sensitivity analysis. Once proven, this result would add additional credibility to the MoGP model presented and help provide better constraints as to what is needed (sample size, training years, visits per patient, etc.) to make the MoGP model best suited for future predictions compared to prior ALS models.

The cluster results for our ALS datasets are stable to the mixing parameter. See below for parameter sensitivity analysis of the prediction task. Four different alpha values were tested: 0.1, 0.5, 2.0, and 5.0 times the original mixing parameter. Both the overall error as well as the number of clusters is reasonably stable across the experiments. In the absence of structure in the data, we would expect the mixing parameter to have a direct effect on the cluster sizes; however, the stability of the results across these parameters points to data structure that is learned in the training process.

The results instead indicate that overfitting likely drives the differences in the model performance. The experiments with fewer training data points are more susceptible to overfitting in the case of a complex model like the flexible gaussian process; however, as more data is provided, the MoGP better captures the data structure. We have included this result and discussion in the supplement.

■ Mixture of Gaussian Processes Model (MoGP)

■ Linear Kernel Model (LKM)

CEFT (Alpha: 0.1x)

CEFT (Alpha: 0.5x)

CEFT (Alpha: 2.0x)

CEFT (Alpha: 10.0x)

CEFT (Alpha: 0.1x)

CEFT (Alpha: 0.5x)

CEFT (Alpha: 2.0x)

CEFT (Alpha: 10.0x)

Supplement Figure 3. Parameter sensitivity analysis showing effect of scaling alpha on the prediction experiments, both for relative error and number of clusters.

6. Most machine learning/AI modeling papers have a model workflow or pipeline figure that clearly articulates the steps of the workflow and/or involved algorithm(s). Such a figure would really help this work. Additionally, have a pseudocode table or figure with more pertinent algorithm details in the supplement would help...particularly for training, parameter tuning, and optimization steps.

Model workflow and pseudocode have been added to the supplement:

Model Workflow

The below section details the specifications for model training, optimization, and parameter initialization.

Methods Supplement Figure 1: Input, training, and optimization visualization

Methods Supplement Table 1: Mixture of Gaussian Processes Workflow

Input: Initial number of clusters, model priors*

Output: Optimized model parameters, latent cluster probabilities

Initialize model parameters and cluster assignments

For 1, ..., N_iterations do

 For 1, ..., N_patients in random order do

 Remove patient from current cluster

 For existing clusters 1, ..., K

 Compute the probability of assigning patient to cluster conditioned on the cluster assignments of all other patients, all other patient trajectories, and model priors.

 Compute the probability of assigning the patient to a new cluster conditioned on the cluster assignments of all other patients, all other patient trajectories, and model priors.

 Sample a new cluster assignment according to the computed cluster assignment probabilities.

 Add patient observation to cluster based on sample

 Remove empty clusters, if needed

*Generative model and priors:

GP Regression:

Signal Variance: Fixed to 1

Lengthscale: Gamma prior with mean 4., variance 9

Mean function slope: Gamma prior with mean 2/3, variance 0.2

Noise variance: Gamma prior with mean 0.75, variance 0.25**2

Threshold: 0.5 for z-score normalized data

Number iterations: 100

7. MINOR: The authors need to revisit the technical language. The use of first person language and pronouns throughout is more in line with an IEEE conference proceeding than a clinical or domain journal.

Based on a sampling of other *Nature Computational Science* articles which also use first person language and pronouns, we currently are keeping this format, but are happy to switch if the editors recommend doing so.

Reviewer #1 (Remarks to the Author: Impact):

The presented MoGP model definitely adds to present discussion in the field that ALS progression is predominantly non-linear. The overwhelming number of non-linear clusters is the most impactful result. However, the paper in its present form, does not compare to enough baselines to illustrate outright ALS progression prediction superiority across the board - in other words, it does not prove it's the best ALS model out there. The clusters are also very interesting in terms of clearly illustrating the preponderance of non-linear progressions, but they fail to fully connect to domain features that a clinical audience will appreciate. The authors do write some text with a couple of cluster examples and how they map to survival; however, more work is needed to make this a key point [if the authors want this to be a key point within their present scope of work]. In summary, focusing more on the non-linear result (which is clearly and quantitatively proven) is the strongest part of the work. However, that result is

somewhat buried in the present text. Restructuring would help emphasize this finding more and minimize some of the less impactful areas where future work is still needed.

We agree with the reviewer that the “overwhelming number of non-linear clusters is the most impactful result” and highlight two important aspects: the number of clusters, which demonstrates the heterogeneity of ALS, and the non-linearity, which is still not sufficiently accounted for in many other approaches. As the reviewer notes, we do not want to make survival prediction a key point. Instead, we include prediction experiments here as an evaluation of cluster robustness. We recommend existing patient-specific models explicitly suited to prediction if a user desires a model that can be applied at diagnosis. We have added a future work section which discusses extensions to connecting clusters to domain features, and updated the section heading following this reviewer’s feedback regarding the structure of the text.

Reviewer #1 (Remarks to the Author: Strength of the claims):

1. The authors should compare the population MoGP model to single patient Gaussian regression models. This is more of an apples-to-apples comparison. The comparison to the linear slope models and such should only be used to emphasize the necessity to model non-linearity.

We have added this model as comparison.

2. If the authors want to make the clusters be central to their work beyond illustrating the number of clusters that were non-linear progression versus linear, more detail and context needs to be given to the clusters' ties to clinical metrics. The few sentences with sparse examples on survival and respiratory function are not enough. OR the authors need to tone the language down on the significance of the clusters to only focus on importance of non-linearity and then write a limitations and future directions section to discuss future mapping of clusters to other clinical variables/features, citing basic examples there.

We appreciate this point; we have added a section on limitations and future directions, in which we note future work to correlate clusters with clinical and molecular features.

Reviewer #1 (Remarks to the Author: Reproducibility):

1. The authors need to provide more information on training and optimization protocols. Pseudocode tables would be helpful context.

We have added this in supplemental methods.

2. While the code will be provided, some more details are necessary in the paper. Also, the authors give no detail in the paper on the types of software packages used, what type of computational environment the model was run on, etc.

We have added this in the code availability section:

Code availability

We provide the python code for the MoGP framework as well as the pre-trained reference model described here for researchers to use to generate predictions of cluster membership and trajectory function from input patient data. We also provide a pip-installable Python package associated with this work (*mogp*). All code used for data processing, modeling, and figure generation can be found at: <https://github.com/fraenkel-lab/mogp>

Python version and specific package versions used for analysis listed below

Python: 3.7.3

Packages:

joblib: 1.0.0
numpy: 1.19.4
pandas: 1.3.1
openpyxl: 3.0.5
sas7bdat: 2.2.3
seaborn: 0.11.1

statannot: 0.2.3
lifelines: 0.25.7
statsmodels: 0.12.2
mogp: 0.1.1
jupyter: 1.0.0
Gpy: 1.9.9
scipy: 1.7.3
scikit-learn: 0.21.1
sklearn: 0.0
matplotlib: 3.1.1

Computational environments: Model run on Azure and compute cluster.

Azure specifications: Standard F32s_v2 machines (32 vCPUs, 64 Gb Mem)

Cluster: 16 cores, 1 node, 10GB memory

Reviewer #2 (Remarks to the Author: Overall significance):

In this study, the authors proposed a new approach to quantify disease progression in ALS. Since linear models are not ideal, the authors explored aggregating patient trajectories in individualized clusters, each with a specific course, regarding rate and curve features.

Overall, this text is not simple to be followed by most neurologists caring ALS patients. To reach greater clinical impact some technical simplification is recommended, if this is the target.

We have worked to simplify the introduction, discussion, and figures for easy clinical interpretation; we have also added a clinically-focused limitations and future work section. We retain some technical details in the results to also ensure utility for a computational audience.

The authors used 4 databases. Three are relatively small, Answer ALS, CEFT and EMORY, regarding the first we are not aware who introduced the data in the site (patients?), concerning the latter, the very fast rate of decline indicates that it represents a quite specific group of patients.

We have specified that “All scores used for this analysis are clinician-reported.” (including AnswerALS, where data was collected by clinicians across eight hospital sites).

We have added the following section under “Limitations” discussing the biases of each dataset:

“Like many clinical studies, the datasets and therefore the progression patterns in this analysis are influenced both by selection bias and attrition bias. Selection bias refers to the sample of the population that is included in each study. Studies like AALS, which require enrollment and consent to undergo additional monitoring, tend to be biased towards slower progressing ALS. The EMORY dataset, which has a high percentage of enrollment from clinic, is likely to be more reflective of a clinical population, although it reflects a group of patients with higher rates of progression on average. Overall though, observational studies tend to have less standardized frequencies of data collection and sparser measurements. On the flip side, clinical trial datasets typically collect extensive longitudinal data, but because of enrollment criteria, can be skewed towards faster progressing individuals. Attrition bias also plays a strong role in ALS datasets, given the rapid pace of disease progression, with patient monitoring becoming increasingly difficult in late-stage disease; this bias may particularly affect the tail end of the identified trajectory patterns. Given the large sample size in our study, and the consistency of the patterns across datasets, we expect that we are sampling the clinical population as broadly as possible, although future work will involve determining the extent to which these trajectories remain consistent in new datasets. It will particularly be interesting to see how trajectory patterns change in the case that emerging ALS medications become clinically approved.”

Two databases (PRO-ACT and CEFT), they partially overlap, which is probably not a good solution regarding training and validation of their model.

We have clarified the text to note that CEFT is not intended as a separate validation set, and amend the text to the following: “We tested this using the largest dataset with sufficient longitudinal measurements: PRO-ACT, which is a large compendium of data from several clinical trials. We also examined the data from CEFT, which is a small clinical cohort within PRO-ACT that may be more representative of common clinical settings.”

Due to data use restrictions, we are not allowed to determine which individuals may overlap between PRO-ACT and CEFT, which is why we cannot separate them into distinct, non-overlapping validation sets.

Other large databases are available, in particular in Europe (Westeneng HJ, et al, Prediction of personalised prognosis in patients with amyotrophic lateral sclerosis: development and validation of a prediction model, Lancet Neurology 2018), which could be used in this study.

We have analyzed an additional dataset, ALS Natural History, which includes 907 individuals and increases the total number of samples in this study to 5161. Like the other datasets, this dataset shows clear nonlinearity, supporting the manuscript's claims. The new figures of the paper are copied below.

Unfortunately, the data from the Westeneng HJ publication is unavailable. We reached out to the corresponding author as well as a number of the European-based site coordinators. We were informed that the data sharing agreements with ENCALS (the model from Westeneng HJ) state that the data we obtained can only be used for the purpose of the prediction model development. They noted that they would have to redo the data sharing agreement in order to use the data for other purposes. In addition, they noted that the data they obtained for their prediction model contains eight covariates and the survival data, and doesn't have longitudinal ALSFRS-R; they asked only for the first ALSFRS-R measurement per patient.

Supplement Figure 1D. Clusters spanning 90% of all individuals in NATHIST visualized. The baseline slope is calculated as the difference between 48 and the mean cluster score one year after symptom onset. N indicates the number of ALS patients in each cluster.

Figure 2. Estimating nonlinearity of trajectories. (A) Cumulative distribution function (CDF) of RMSE between a participant's predicted cluster membership and cluster model mean. P-values calculated with Kolmogorov-Smirnov two-sample tests between MoGP and LKM distributions, and between MoGP and SM distributions (B) A subset of nonlinear clusters from PRO-ACT visualized; N indicates number of ALS patients per cluster.

Figure 4. Assessing trajectory consistency across datasets. A) The reference model was trained on PRO-ACT and used to predict progression trajectories of participants in other datasets; the four largest reference model clusters are shown. B) Average test error between cluster mean function and participant ALSFRS-R scores, using the reference model and study-specific models. * p -val < 0.05 (Wilcoxon signed-rank two-sided test). Error bars show 0.95 confidence interval between 5 splits.

For this model the authors do not mention the number of required patients for its development.

Dirichlet processes allow for the identified number of clusters to scale relative to the size of the dataset. In particular, the reference model can be used with clinical measurements from any number of patients, including one.

We have added the following to the discussions text: “One of the properties of the Dirichlet process model underlying MoGP clustering is that it will naturally scale the number of identified clusters within a given dataset depending on the number of samples in that dataset; in particular, one can use a reference model on a clinical cohort of any size. This non-parametric property of the model underlies the difference in total number of clusters found in the varying datasets.”

Exclusion criteria are very loose and arbitrary. Patients with a first visit more than 24 or 36 months after disease onset would not be accepted in a trial (the authors propose that their tool could be used in future clinical trials), they decided for 7 years; and improvement of ALSFRS-R greater 6 points is never observed in an ALS clinic (if the diagnosis is correct), why not 2 or 3, considering and acceptable fluctuation? Did they include patients with PEG or NIV at entry?

Because we are proposing the use of a data-driven model, we aimed to be as conservative as possible in removing patients from the dataset so as to not introduce additional selection bias. 7 years was selected as the point in which longitudinal data became sparse; this is expected given the average duration of ALS survival. 6 ALSFRS-R points were recommended in consultation with our clinician collaborators, because, like the reviewer mentions, a jump like this was unlikely to be seen unless there was a data-entry error. The monotonic inductive bias in the model helps adjust for fluctuations smaller than this. We have added a justification of the exclusion criteria to the methods.

Patients with PEG (gastrostomy; measured in ALSFRS-R 5) and NIV were not explicitly excluded from the model. It is important to note that these interventions, though, do affect ALSFRS-R scores. We have added a sentence to the introduction reflecting this limitation of the ALSFRS-R metric: "Furthermore, interventions like percutaneous endoscopic gastrostomy (PEG) and non-invasive ventilation (NIV) can affect these clinical metrics."

Results are good. Regarding survival, it would be convenient to compare outcome of their model with the ones published applying different models (Westeneng HJ, et al, Prediction of personalised prognosis in patients with amyotrophic lateral sclerosis: development and validation of a prediction model, Lancet Neurology 2018).

Our model does not explicitly predict survival and differs fundamentally from the ENCALIS (Westeneng HJ) model. The ENCALIS model aims to predict time to survival using a limited set of clinical covariates available at baseline; it does not consider longitudinal ALSFRS-R. Our model aims to predict changes in the ALSFRS-R score over time. While ALSFRS-R severity correlates with survival duration, the correlation is confounded by a number of clinical interventions; the two models accomplish significantly different tasks.

We have, though, added additional baseline models to our analysis (see above; Reviewer 1: #3). We also previously provide comparison to the patient-specific sigmoidal model (D50), and recommend this model if users want a patient-specific estimate as opposed to identification of clusters.

Regarding the sigmoidal model, we note the following in the results:

"Since there are many settings in which patient-specific parametric models are very useful, we compared our model with a patient-specific sigmoidal model (SG).²⁰ Somewhat surprisingly, despite the fact that the MoGP models groups of patients, rather than individuals, MoGP outperforms a patient-specific sigmoid model by one or more ALSFRS-R points for 4.20%-9.43% of patients across the studies (Supplementary Table 4). This indicates that while a sigmoidal model captures much of the non-linearity, it does not represent the full complexity of progression patterns."

Some results using FVC predicted value and ALSFRS-R subscore were mentioned in results (see figures), but there no relevant information in methods about these analyses.

The following was added to the methods: "We also trained MoGP models on forced vital capacity percentages (calculated as the maximum of three trials) and ALSFRS-R subscores (fine motor, gross motor, respiratory, and bulbar domains), and evaluated trajectory patterns. A maximum score of 100% was used for the forced vital capacity percentage model, and a maximum score of 12 was used for ALSFRS-R subscores."

Discussion is appreciated.

Minor

Last paragraph of the Introduction summarized the article, which is not necessary.

Based on a sampling of other *Nature Computational Science* articles which also include a summary as the last paragraph of the introduction, we currently are keeping the paragraph, but happy to remove it if the editors recommend doing so.

The text is somewhat repetitive in some parts, for example last paragraph on page 4 is replicated in Modelling Approach on the next page, and PRO-ACT features are described on pages 9 and 10.

We have corrected this.

Reviewer #2 (Remarks to the Author: Impact):

This is a good work, with potential great impact. Possibly *Nature Computational Science* would be the best room.

We thank the reviewer for their comment

Reviewer #2 (Remarks to the Author: Strength of the claims):

ALS is a very competitive area, and computational modelling is a new exciting field. After revision, this manuscript has a great chance of a relevant impact. To use another large data base in addition to PRO-ACT would strengthen their conclusions, they used 3 other relatively small, and one with overlapping with PRO-ACT.

We have added the ALS Natural History study (N=907) to the analysis.

Reviewer #2 (Remarks to the Author: Reproducibility):

I believe this could be reproduced by other authors.

Reviewer #3 (Remarks to the Author: Overall significance):

This study provides a characterisation of the longitudinal trajectory of the ALSFRS-R in amyotrophic lateral sclerosis. The developed model was also validated in other datasets. The result is original and can be applied to other fields where longitudinal data is available and behaves non-linear.

Reviewer #3 (Remarks to the Author: Impact):

Because of the complexity of the model (which a personally appreciate and that is explained and investigated well by the authors) I have some doubts about the implementation in practice.

Regarding ease of implementation - we have made an effort to provide two documented tutorials (<https://github.com/fraenkel-lab/mogp/tree/main/example>), which show a user how to train a model, and use the reference model provided here.

Reviewer #3 (Remarks to the Author: Strength of the claims):

Ramamoorthy et al. studied the longitudinal trajectory of the revised version of the amyotrophic lateral sclerosis functional rating scale (ALSFRS-R). The authors developed and validated a sophisticated Bayesian non-linear model for the longitudinal trajectory of the ALSFRS-R. Reviewing this well-performed study was a great pleasure but I have also some comments aiming to further improve this study.

1. The authors frame their study as 'modelling ALS progression'. ALS progression is, however, much broader than patients daily functioning which is measured by the ALSFRS-R. It would be great if the authors could be clearer about this throughout the abstract and manuscript.

We appreciate this point, and have clarified this throughout the text.

We have added the following sentence to the abstract: "We focus on a clinical definition of disease progression that reflects changes in patient function, as measured by the revised ALS functional rating scale (ALSFRS-R) or forced vital capacity."

We have also added the following to the introduction: "These clinical metrics, such as the Revised ALS Functional Rating Scale (ALSFRS-R), are a proxy for disease progression in ALS, typically measuring patients' daily function."

2. I agree with the authors that characterizing heterogeneity in ALS is important but the last sentence of the abstract 'Our results provide a critical advance in characterizing the heterogeneity in disease progression patterns of' is somewhat overstated. This also applies to the last sentence of the introduction."

We have modified the last sentence to reflect this: "Our results advance the characterization of disease progression patterns of ALS."

3. Minor. In paragraph 2 of the introduction, the authors discuss the change in ALSFRS-R slope that is used in clinical trials. They classify ~0.4 points difference as a small effect, but given that the average decline of the ALSFRS-R is 0.5-1.0 points per month in population-based datasets (in trial populations it might be somewhat higher) this needs to be adjusted. Moreover, edaravone is not approved in Europe.

We have removed the characterization of 0.4 as a small effect, instead noting that “Improvements in the linear rate of decline of the ALSFRS-R are assumed to correspond with clinically meaningful efficacy.”

We have clarified that edaravone was only approved in the US: “For example, edaravone was approved in the United States based on a 2.5 ALSFRS-R point difference in decline between the treatment and control arms over 6 months¹⁴”

4. Minor. In the first sentence of the fourth paragraph of the introduction a typo might have occurred (“the more a more”).

We have corrected this.

5. Table 1. The distribution of ‘number of visits’ and ‘months followed’ can be very skewed. A median and range (or interquartile range) would be more appropriate. It is unclear how the ALSFRS-R slope was calculated in this table. This is important because the degree of decline can be very skewed and, if possible, a more robust measure of this slope would be preferred over the mean and standard deviation. Finally, the number of characteristics provided is too little to be sufficiently informed about the datasets used. In summary, more detail is needed.

We have updated Table 1 and Supp. Table 1 with median and IQR values. The text in the “Study Populations” has also been updated to reflect median summary statistics. We have also added to the legend of Table 1: “Slope is calculated using linear regression for all measured data per individual, in points per month.”

We have included a table (Supplement Table 1, copied below) with additional available clinical characteristics such as sex, limb and bulbar onset, and age of onset. Unfortunately, for some studies, our data sharing was restricted to a limited set of clinical features, so features beyond the below table would be difficult to collect at this time; in our text, we cite the datasets for each study where further details can be found.

	No. (%) Male	No. (%) Female	No. (%) Limb Onset	No. (%) Bulbar Onset	Median (IQR) Age of Onset	Total No. Participants Included
PRO-ACT	1838 (62.9)	1085 (37.1)	2001 (68.5)	589 (20.2)	55.10 (16.11)	2923
NATHIST	537 (59.2)	368 (40.6)	621 (68.5)	239 (26.4)	61.73 (14.24)	907
CEFT	289 (60.7)	187 (39.3)	377 (79.2)	108 (22.7)	54.70 (15.12)	476
AALS	287 (62.9)	169 (37.1)	344 (75.4)	111 (24.3)	57.85 (13.96)	456
EMORY	233 (58.4)	166 (41.6)	N/A	N/A	61.09 (16.40)	399

Supplement Table 1: Study Populations – Extended Summary Statistics. IQR indicates interquartile range. N/A indicates not reported.

6. Around 3000 participants from the PRO-ACT database were included (which was by far the largest dataset used). Can the authors please comment on this selection? Which criteria were used to select these patients from the PRO-ACT database and why? What happened when more patients from the PRO-ACT database were included?

While PRO-ACT itself is quite large, only 3,264 individuals had recorded revised ALSFRS-R scores. A number of the remaining clinical trials in PRO-ACT only measured ALSFRS without the respiratory domain or had incomplete data; for the purpose of this analysis, we chose to only include studies which had complete ALSFRS-R scores because it is the current standard of the field.

Of the 3,264 individuals with recorded completed ALSFRS-R scores in PRO-ACT, 2990 had three or more ALSFRS-R visits. 2923 individuals met all remaining criteria (see below) and were included in the model; this represents a high rate of model inclusion - 90% of all available individuals with ALSFRS-R in PRO-ACT.

We have added details and justification to the methods on the criteria for our exclusion criteria in the methods: "Because we are proposing the use of a data-driven model, we aimed to be as conservative as possible in removing patients from the dataset so as to not introduce additional selection bias. For this analysis, participants were excluded from the model if fewer than three complete ALSFRS-R visits were recorded, the first visit was more than seven years from symptom onset, or an increase of greater than six points in ALSFRS-R between subsequent visits was recorded (Table 1). Seven years was selected as the point in which longitudinal data became sparse. Six ALSFRS-R points was selected because a jump like this was unlikely to be seen unless there was a data-entry error."

7. Subjects with at least 4 visits were used for prediction and subjects with at least 10 visits were used for assessing interpolation. Could the authors please provide analyses of what happened when fewer data points were available (to really demonstrate how robust each model is to sparse data)? This is even more important because (even from a trial population with usually a lot of measurements, i.e. PRO-ACT) >50% of the subjects were excluded because they have less than the data points needed. And could the authors please evaluate not only the MoGP, SM and LKM model but also the sigmoidal model? And which results were obtained when performing these analyses in the other datasets?

For our sparsity tests (Figure 3A), the median number of visits for the first set of bars (25% included data) is 2 visits. When only 2 visits are included, the slope model outperforms MoGP. The median number of visits for 50% is 4 visits; at this threshold, MoGP outperforms the slope model.

The sparsity and prediction tasks are intentionally set up with more data in order to test the extent to which the model can recapitulate those data points. For the sparsity task, at the sparsest experiment, 25% of the 10 visits were provided, which meant that 2 visits per individuals were shown to the model; because the model requires longitudinal visits, going below this number of visits would mean that we would not have enough data to appropriately test the inclusion of 25%, 50%, and 75% of the data. We acknowledge that this experimental setup can potentially cause the bias of excluding the fastest progressing ALS (reflected in the summary statistics in Supp Table 1).

We have added additional baseline models to our analysis (see above; Reviewer 1: #3). We also previously provide comparison to the patient-specific sigmoidal model (D50), and recommend this model if users want a patient-specific estimate as opposed to identification of clusters.

Regarding the sigmoidal model, we previously noted the following in the results:

"Since there are many settings in which patient-specific parametric models are very useful, we compared our model with a patient-specific sigmoidal model (SG).²⁰ Somewhat surprisingly, despite the fact that the MoGP models groups of patients, rather than individuals, MoGP outperforms a patient-specific sigmoid model by one or more ALSFRS-R points for 4.20%-9.43% of patients across the studies (Supplementary Table 4). This indicates that while a sigmoidal model captures much of the non-linearity, it does not represent the full complexity of progression patterns."

Below are the interpolation and prediction experiments for NATHIST. While AALS and EMORY unfortunately do not have enough longitudinal data for the sparsity experiment, we were able to conduct the prediction experiment, with results below:

Supplement Figure 5. Prediction and interpolation results on additional datasets. *p-val $\leq 1e-1$ **p-val $\leq 1e-2$ ***p-val $\leq 1e-3$ ****p-val $\leq 1e-4$ (Wilcoxon signed-rank one-sided test). Error bars show 0.95 confidence interval.

These datasets all have fewer longitudinal visits than the PRO-ACT results used in the text, and therefore significantly less statistical power. The results on the AALS datasets are not statistically significant. The EMORY dataset shows statistically significant improvements of the MoGP over LKM starting at 1.0 years. NATHIST prediction is statistically significant at 2.0 years, although shows a similar trend as the other datasets starting at 0.5 years. NATHIST interpolation shows a statistically significant improvement over LKM at all thresholds. The overall trends in these datasets matches that in the original text, specifically that at the sparsest cases, a linear model is more appropriate than MoGP, but with additional longitudinal training data, the MoGP outperforms the LKM model.

8. When reading the methods section about ‘Model Generalizability’ I first interpreted that the model developed in the PRO-ACT database (i.e. reference model) was modified before it was applied to other datasets. After reading the results it became clear that this was not the case. Could the authors please clarify that the primary analysis was to develop a model in the PRO-ACT database and apply this unchanged to the other datasets? The ‘study specific models’ can be mentioned as additional sensitivity analyses to investigate possible overfitting of the reference model. And as a minor comment, it could be added that the test and train datasets were split randomly.

We have added the following to the methods:

“The reference model was not modified in any way prior to being applied to other datasets.”

“The study-specific models allow us to evaluate possible overfitting of the reference model; an error on the reference model that is significantly higher than that of the study specific models indicates poor model generalizability.”

“We split all of our study populations into test and training datasets (60% train, 40% test; repeated across 5 randomly split test/train datasets).”

9. From the methods it was somewhat unclear how the ‘monotonic inductive bias’ was incorporated. After reading the supplement this became clear for the ‘negative linear mean function’, but I still have some difficulties with interpreting the ‘threshold function’. Could the authors please consider improving the section about ‘monotonic inductive bias’?

To the main methods text, we have added: “To further encourage declining trajectories, we modify the Dirichlet process clustering algorithm, such that an individual can only be placed in a cluster if their score at their initial visit is not significantly higher than the mean function of the current cluster at that point.”

For the supplement, we have added the following details regarding threshold function: “To encourage monotonically declining functions, we use two modifications to MoGP: 1) a negative linear mean function in our GPs, and 2) a thresholding function to determine cluster membership. In our sampling procedure for our DP model, the probability of each individual joining each cluster is calculated. Our thresholding function constrains the number of clusters an individual can join. If the score for initial visit for a given sample is not close (where close is defined by a user-set ‘threshold’ parameter) to a cluster’s mean function, then the algorithm sets the probability of joining that cluster to 0. This prevents the probability that a participant with a starting ALSFRS-R score vastly divergent from a given cluster will be added to the cluster. For these experiments, this threshold is set as 0.5 for z-scored data, which roughly approximates 5 ALSFRS-R points, because it would be clinically unlikely that one sees this large of a range in patient function for a trajectory pattern.”

10. The authors claim: ‘The heterogeneity of the populations enabled us to measure the robustness of our model to data collection methods and the generalizability of ALS progression patterns between varying study populations’. This claim about heterogeneity is very much dependent on the underlying causes of this heterogeneity and not so much on the few measures provided (ALSFRS-R slope and follow-up duration). This refers also to my comment about table 1.

We have specified the causes of heterogeneity we are referring to, now saying: “The differences between the populations allow us to measure the robustness of our model to data collection methods, frequency of clinical visits, and duration of follow-up; together, these results point to the generalizability of ALS progression patterns between both clinical trial and observational study populations.”

11. In supplementary figure 1 different clusters of the non-PRO-ACT dataset are plotted. These figures, however, display only a relatively small part of the data: AALS 284 of 456 patients (62%), CEFT 216 of 476 (45%) and EMORY 282 of 399 (71%). Could the authors please show an increased number of figures to demonstrate the different clusters? I would suggest that clusters of at least 90% of the data would be provided. This also applies to the clusters found in the PRO-ACT database (figure 1) which includes 1573 out of 2923 patients (54%). If Figure 1 becomes too large it possibly can be provided supplementary. Moreover, different clusters look very similar. Can the authors please provide a similarity score between clusters?

Extended versions of the figures covering 90% of data in Supplementary Figures 1A-E have been provided.

We have also added analysis of similarity between clusters, providing two metrics (length scale and mean function) with which to score similarity. We have also analyzed the structure that emerges when our MoGP clusters are themselves clustered using these two metrics (Supp Fig. 7)

We have added the below text regarding cluster similarity scores:

“Another interesting property of these clusters is their variation both among the rate of progression and the stability of their progression patterns. MoGP enables the characterization of each of these properties through the mean function slope and kernel function lengthscale parameters respectively, both of which are learned and optimized through the training process. These properties allow for the sub-selection of clusters with desired properties. For example, to only analyze fast-progressing individuals, a user can select clusters that have a high slope. The lengthscale property indicates a function’s stability over time; analysis which rely on assumptions of linearity may benefit from selecting clusters with higher lengthscales. The model provides scores for each of these parameters, and these can be used to approximate similarity between clusters depending on the desired clustering property (Supp. Fig. 7).

Clustering trajectories based on the optimized slope and lengthscale parameters reveals interesting patterns (Supp. Fig. 7). This analysis highlights the dominant progression patterns in ALS: sigmoidal fast progression (Supp. Fig 7B, 17.48% of individuals),

stable slow progression (Supp. Fig 7E, 17.38%), unstable slow progression (Supp. Fig 7F, 32.98%), and unstable medium progression (Supp. Fig 7D, 30.82%). Clusters following a sigmoidal fast progression pattern have the highest percentage of individuals with bulbar onset (30.14% of individuals), while those in the stable slow progression pattern have the highest percentage of individuals with limb onset (76.97%) (Supp. Table 5). The median ages of onset for the patterns ranges between 54 to 59 years (Supp. Table 5).”

Supplement Figure 7. Dominant ALS progression patterns, identified using lengthscale and negative mean function slope. Lengthscale indicates trajectory stability; negative mean function slope corresponds to rate of progression. Learned model parameters from the PRO-ACT reference model are k-means clustered (Left plot; k=6, marker size corresponds to cluster size), with clusters $\geq N=5$ visualized, and percentage of individuals that fall within each of the trajectory patterns labeled (Right plots).

12. 92 clusters were found using the PRO-ACT data, while in the other datasets a maximum of 34 clusters was found. This is intrinsically related to the methods used but the meaning of this difference needs to be discussed in the discussion.

We have added the following to the discussions text: “One of the benefits of the Dirichlet process algorithm underlying MoGP clustering is that it will naturally scale the number of identified clusters within a given dataset depending on the number of samples in that dataset; in particular, one can use a reference model on a clinical cohort of any size. This non-parametric property of the model underlies the difference in total number of clusters found in the varying datasets.”

13. Minor. Based on the text, the ‘greater than sign’ in supplementary tables 2, 3 and 4 should be replaced by a ‘greater than or equal to sign’.

The sign should be ‘greater than’. We have corrected the text to read: ‘Across the populations, using the MoGP results in a lowered error of greater than one ALSFRS-R point as compared to the LKM for at least 27.16% of participants; at least 8.33% of patients have an improvement in accuracy greater than two ALSFRS-R points (Supplementary Table 2).’

14. Minor. I think that figure 2a and 2b have different messages and it might result in interpretation difficulties to combine them. Maybe these figures can be split up into separate figures?

For now, we have decided to keep these together, with figure 2a intending to quantitatively demonstrate nonlinearity, and figure 2b intending to visualize nonlinear trajectory patterns.

15. Figure 3. These figures are now somewhat difficult to read, especially the error bars (which are very small). I think that it would be much more straightforward to interpret when the authors plot the full distribution of the (absolute) deviation of points from the modelled mean (if needed with a log or square root transformation). Some readers might for example interpret the first blue bar in Figure 3A as an error of 3 points (with very small error bars) which could be interpreted by some readers as that the model has nearly always an error of 3 points, which is a lot. Significant differences with other models have only a very limited meaning because all these models perform suboptimally. This interpretation can be prevented by just plotting the absolute differences and it provides also more insight into the full distributions of errors.

We appreciate this comment. Below is the requested plot of the root mean square error with visualized data points. Unfortunately, the hundreds of data points per bar make it difficult to visualize the full distribution; while the vast majority of

samples have low errors, a few outliers have high errors, making the plot difficult to read. We have added this plot to the supplement, and also provide the full data underlying the plot as a supplementary file.

Supplement Figure 5. Visualizing full distribution of error for interpolation and prediction. Box plot represents interquartile range; whiskers indicate proportion (1.5) the IQR past the low and high quartiles to extend the plot whiskers. Points outside the whisker range represent outlier samples.

16. Minor. In the supplement, the authors describe the α parameter, which indicates the scaling parameter of the beta prior, but no value for this parameter is provided.

We have added the following to the supplement:

“Alpha indicates the scaling parameter and modifying this can influence the degree of cluster discretization and therefore the number of identified clusters. For these experiments, to encourage large clusters with at least 50 individuals per cluster, alpha was set to the number of patients in a given dataset divided by 50.

We also show a parameter sensitivity of analysis of the model’s relative error and number of clusters (Supp. Fig. 3, see Rev. 1 #5). Four different alpha values were tested: 0.1, 0.5, 2.0, and 5.0 times the original mixing parameter. Both the overall error as well as the number of clusters is reasonably stable across the experiments. In the absence of structure in the data, we would expect the mixing parameter to have a direct effect on the cluster sizes; however, the stability of the results across these parameters points to data structure that is learned in the training process.”

17. I could not find a discussion of possible limitations. Besides my suggestions above, I think that the lack of population-based datasets can be seen as a potential limitation and could lead to selection bias. Furthermore, attrition bias is a common problem in ALS research. Could the authors please discuss these two biases that could be present in their study and what the meaning of these biases is for the interpretation of their study?

We have added a limitations section after the discussion, which includes the following text: “Like many clinical studies, the datasets and therefore the progression patterns in this analysis are influenced both by selection bias and attrition bias. Selection bias refers to the sample of the population that is included in each study. Studies like AALS, which require enrollment and consent to undergo additional monitoring, tend to be biased towards slower progressing ALS. The EMORY dataset, which has a high percentage of enrollment from clinic, is likely to be more reflective of a clinical population, although it reflects a group of patients with higher rates of progression on average. Overall though, observational studies tend to have less standardized frequencies of data collection and sparser measurements. On the flip side, clinical trial datasets typically collect extensive longitudinal data, but because of enrollment criteria, can be skewed towards faster progressing individuals. Attrition bias also plays a strong role in ALS datasets, given the rapid pace of disease progression, with patient monitoring becoming increasingly difficult in late-stage disease; this bias may particularly affect the tail end of the identified trajectory patterns. Given the large sample size in our study, and the

consistency of the patterns across datasets, we expect that we are sampling the clinical population as broadly as possible, although future work will involve determining the extent to which these trajectories remain consistent in new datasets. It will particularly be interesting to see how trajectory patterns change in the case that emerging ALS medications become clinically approved.”

Reviewer #3 (Remarks to the Author: Reproducibility):

The analyses were appropriate. The developed model was validated in other datasets. The code for this study is provided online.

Comments from editors:

Reviewer #1 has mentioned that the authors have over-represented the clinical significance of their clusters and they should mention similar use cases of MoGP in other diseases. They have also indicated the need for assessing the population-level MoGP generalizability as compared to the Gaussian models for each individual patient.

We have added a limitations section addressing future work to connect clusters to additional features within the clinical domain. We have added discussion of similar use cases of related GP models in other diseases, and added an individual-patient GP model as comparison.

Reviewer #2 has requested technical simplification and comparing the outcome of their model previously published work.

We have edited the introduction, discussion, and limitations/future work with details relevant to a clinical audience; we have kept some technical components of the results at this point to also be amenable to a computational audience. We have also added additional baseline models to our analysis.

Reviewer #3 has mentioned that the claims are overstated and more details are needed about the datasets used in the paper along with implementation on fewer data points to test the robustness.

We have drawn back claims and added a future work and limitations section. We now provide additional details about the datasets used in the paper. We extend experiments showing the effect of fewer data points to additional datasets.

Points to be addressed for *Nature Computational Science* (NCS): The editors at NCS will need to see addressed all points raised by the reviewers in full.

Points to be addressed for *Nature Communications*: The editors at *Nature Communications* think that the advance provided by the study is sufficient for their journal. The editors at *Nature Communications* will need to see addressed all points raised by the reviewers in full.

Points to be addressed for *Communications Biology*: The editors at *Communications Biology* will need to see addressed all points raised by the reviewers in full.

Nature CS: Major revisions with extension of the work

The editors at NCS think that, despite the computational framework not being novel in a broad sense, the work could have a positive impact in the ALS field. They will need to see addressed all points raised by the reviewers in full. Additionally, they would like to see qualitative or quantitative discussion on how the proposed method can be applied in domains different from the ALS one.

We have applied the model in both a Parkinson’s and Alzheimer’s context, and added the below section on this to our text.

Methods:

We applied our method to the Alzheimer’s Disease (AD) Assessment Scale-Cognitive Subscale (ADAS-Cog-13^{42,43}) from the Alzheimer’s Disease Neuroimaging Initiative (ADNI⁴⁴). Individuals with a confirmed AD diagnosis at any point of the data collection were included in the model; this also included individuals who began the study with mild cognitive impairment (MCI) and then converted to AD diagnosis. To ensure sufficient longitudinal data, individuals were fewer than three longitudinal visits were

excluded, with a total of 331 individuals included in the model. The correlation between the learned clusters and AD to MCI conversion was then calculated.

We also applied our method to the Movement Disorder Society-Unified Rating Scale⁴⁵ (MDS-UPDRS) scale from the Parkinson's Progression Markers Initiative dataset⁴⁶ (PPMI). In contrast to ALS and AD, for PD there are medications that can mitigate symptoms but not long-term progression of the disease.⁴⁷ Because we were interested in characterizing progression patterns when not affected by medications, we focused on measurements of the MDS-UPDRS Part III, in the "off state", which is defined as either prior to the initiation of medication or after abstaining from medication for at least 12 hours. Individuals with fewer than three longitudinal off-medication scores or a first visit greater than ten years from symptom onset were excluded, with a total of 397 individuals included in the model. We calculated the correlation between the clusters and PD subtypes of PIGD (postural instability/gait difficulty) and tremor dominant (TD), with the designation of PIGD/TD calculated following the method previously described in Stebbins, et al.⁴⁸ For the purpose of analyzing the PD subtype correlation with cluster membership, we focus on individuals with a stable PIGD/TD designation (one that does not change over the course of the disease).

Results:

The MoGP approach can be applied to functional rating scales that are widely used in other neurodegenerative diseases. We applied MoGP to the Alzheimer's Disease Assessment Scale-Cognitive Subscale (ADAS-Cog-13^{42,43}). The model showed a range of disease progression patterns, with varying severities of progression (Fig. 7A, Supp. Fig. 9). The majority of the largest clusters showed linear trajectories, in which the baseline slope within the first year of data collection appropriately captured later progression; clusters E and H, while largely linear, deviate from baseline slope, showing counter-examples to this trend. Furthermore, the clusters corresponded to varying rates of MCI to AD conversion, with cluster F having the highest number of individuals with an MCI diagnosis at baseline (90.00%), and cluster G having the lowest (5.26%) (Supp. Table 6).

Similarly to ALS and AD, Parkinson's disease (PD) is heterogeneous in its symptom presentation and progression which creates challenges in therapeutic discovery. Unlike Amyotrophic Lateral Sclerosis and Alzheimer's disease, there are widely used medications for Parkinson's disease that can provide symptomatic relief, although do not slow or stop the progression of PD.⁴⁷ We characterized patterns in motor decline by applying MoGP to Part III of MDS-UPDRS scale⁴⁵ using only data from the "off state", i.e., when not affected by medications. MoGP identified a number of progression trajectories (Fig. 7B), with some showing stability of motor scores (pattern C, F), while others showed clear motor function decline (pattern A, B, D). When compared with PD subtypes of PIGD (postural instability/gait difficulty) and tremor dominant (TD)⁴⁸, clusters with an unstable slow progression pattern (Supp. Fig. 10F) have the highest percentage of TD subtypes (91.89%, Supp. Table 7). In contrast with previous studies of the linearity of MDS-UPDRS scores⁵³, our results also point to non-linear complexity in some clusters (clusters C, E, G). While in this study, we have predominantly focused on the utility of our model in the context of ALS, MoGP's flexibility enables it to characterize long-term heterogeneity in time-series metrics in a number of diverse clinical settings.

Nature Communications: The editors at Nature Communications think that the advance provided by the study is sufficient for their journal. The editors at Nature Communications will need to see addressed all points raised by the reviewers in full.

Communications Biology: The editors at *Communications Biology* also think that the advance provided by the study is sufficient for their journal. The editors at *Communications Biology* require that all of the points raised by the reviewers are addressed as much as is feasibly possible and any caveats or limitations are clearly discussed where additional data cannot be provided.

Editorial recommendations:

Our top recommendation is to revise and resubmit your manuscript to *Nature Computational Science*. We feel the additional experiments required are reasonable and in addition, we would like to see applications of the proposed methodology to multiple domains to establish the broad applicability of the study.

You may also choose to revise and resubmit your manuscript to *Nature Communications*. This option might be best if the requested experimental revisions are not possible/feasible at this time.

Data availability statement:

Thank you for including a Data Availability statement. However, we noted that you have only indicated that data are available upon request. The data availability statement must make the conditions of access to the "minimum dataset" that are necessary to interpret, verify and extend the research in the article, transparent to readers.

We now specify conditions of access:

AALS (ClinicalTrials.gov Identifier: NCT02574390) is available for download in the AnswerALS data portal (data.answerals.org). PRO-ACT can be downloaded from the PRO-ACT database (<https://nctu.partners.org/ProACT>). CEFT (ClinicalTrials.gov Identifier: NCT00349622) can be downloaded from National Institute of Neurological Disorders and Stroke (NINDS) (<https://www.ninds.nih.gov/Current-Research/Research-Funded-NINDS/Clinical-Research/Archived-Clinical-Research-Datasets>). EMORY is restricted access at this time due to containing information that could compromise patient privacy, but available with permission from Dr. Jonathan Glass upon reasonable request. NATHIST is available from the ALS/MND Natural History Consortium; please, use the following link: <https://www.data4cures.org/requestingdata> to provide a summary of proposed data use, data elements requested, and publication intent.

All source data underlying the graphs and charts presented in the main figures must be made available as Supplementary Data (in Excel or text format) or via a generalist repository (eg, Figshare or Dryad). This is mandatory for publication in a Nature Portfolio journal, but is also best practice for publication in any venue.

The following figures require associated source data: Figures 1 to 6

Source data is now provided. As a note, for MoGP figures 1, 6, 7, we are unable to provide the raw clinical data due to data use restrictions, but we note how readers can download this data in our Data Availability Statement, and we provide all code to recreate figures from the data.

Data citation:

Please cite (within the main reference list) any datasets stored in external repositories that are mentioned within their manuscript. For previously published datasets, we ask that you cite both the related research article(s) and the datasets themselves. For more information on how to cite datasets in submitted manuscripts, please see our data availability statements and data citations policy: <https://www.nature.com/documents/nr-data-availability-statements-data-citations.pdf>

We have updated citations to datasets for available previously published datasets (CEFT and PROACT).

Code availability and citation

Please include a statement under the heading "Code Availability", indicating whether and how the custom code/software reported in your study can be accessed, including any restrictions to access. This section should also include information on the versions of any software used, if relevant, and any specific variables or parameters used to generate, test, or process the current dataset. Code availability statements should be provided as a separate section after the Data Availability section.

We have added this to the Code Availability section.

Ethics:

Please provide a 'Competing interests' statement using one of the following standard sentences:

1. The authors declare the following competing interests: [specify competing interests] 2. The authors declare no competing interests.

See our competing interests policy for further information:

<https://www.nature.com/nature-research/editorial-policies/competing-interests>

We have amended our competing interests' section to reflect this.

Because your study includes human participants, confirmation that all relevant ethical regulations were followed is needed, and that informed consent was obtained. This must be stated in the Methods section, including the name of the board and institution that approved the study protocol.

We have added the following to the supplemental methods:

Explicit approval was received for all clinical datasets used in the present work. For AALS, the study was approved by local institutional review boards, and all participants provided written informed consent. Consent was uniform across all sites and included agreement to share data broadly for medical research. We received approval for CEFT from the National Institute of Neurological Disorders and Stroke (NINDS). For the original CEFT study, institutional review board approval was obtained at each center, as well as the MGH coordination center IRB, and participants provided written informed consent before screening. We received approval for PRO-ACT from the Pooled Resource Open-Access ALS Clinical Trials Consortium. PRO-ACT is an anonymized database that includes merged datasets from multiple ALS clinical trials. It requires an application to request access, in which the user must agree to protect the security of the data. Dr. Jonathan Glass provided approval and access for using the EMORY dataset. For the original EMORY dataset, the Emory institutional review board approved the study. For NATHIST, each individual site had local IRB approval.

The use of colored axes and labels should be avoided. Please avoid the use of red/green color contrasts, as these may be difficult to interpret for colorblind readers.

We have updated figures to minimize red/green contrasts.

To improve reproducibility of your analyses, please provide details regarding your treatment of outliers.

We have provided additional details on inclusion/exclusion criteria for outliers in our dataset.

The quality of some of the figures appears to be quite low. If possible, we suggest replacing these with higher-resolution images.

We now provide an additional high-resolution copy of each figure.

Decision Letter, Reviewer comments, second version:

Subject: Final revisions for manuscript GUIDEDOA-21-00225A

Message: *Please ensure you delete the link to your author homepage in this e-mail if you wish to forward it to your co-authors.

Dear Professor Fraenkel,

Thank you for your patience as we have prepared the guidelines for final submission of your manuscript entitled "Identifying Patterns of ALS Progression from Sparse Longitudinal Data" for publication in Nature Computational Science as part of the Guided Open Access initiative.

Please carefully follow the step-by-step instructions provided in the attached file, and add a response in each row of the table to indicate the changes that you have made. Please also check and comment on any additional marked-up edits we have proposed within the text. Addressing each point will help to ensure that your revised manuscript can be swiftly handed over to our production team.

When you upload your final materials, please include a point-by-point response to any remaining reviewer comments, which are appended to this email.

If you have not done so already, please alert us to any related manuscripts from your group that are under consideration or in press at other journals, or are being written up for submission to other journals; please see <[a href="https://www.nature.com/nature-research/editorial-policies/plagiarism#policy-on-duplicate-publication"](https://www.nature.com/nature-research/editorial-policies/plagiarism#policy-on-duplicate-publication)>our editorial policies for details.

Policies

In recognition of the time and expertise our reviewers provide to the Guided Open Access process, we formally acknowledge their contribution to the external peer review of articles published in the journal. All peer-reviewed content will carry an anonymous statement of peer reviewer acknowledgement, and for those reviewers who give their consent, we will publish their names alongside the published article.

Guided Open Access also offers a Transparent Peer Review option for all papers published part of this trial. As such, we encourage our authors to support increased transparency into the peer review process by agreeing to have the reviewer comments, author rebuttal letters, and Editorial Assessment report published as a Supplementary item. When you submit your final files please clearly state in your cover letter whether or not you would like to

participate in this initiative. Please note that failure to state your preference will result in delays in accepting your manuscript for publication.

ORCID

Non-corresponding authors do not have to link their ORCID but are encouraged to do so. Please note that it will not be possible to add/modify ORCIDs at proof. Thus, please let your co-authors know that if they wish to have their ORCID added to the paper they must follow the procedure described here prior to acceptance.

Cover suggestions

Open Access

As we discussed in our previous email, all articles published via Guided Open Access are made freely accessible upon publication under a CC BY license (Creative Commons Attribution 4.0 International License). One of our editorial assistants should have already been in touch with you to request the forms necessary for publication. If you have not received this email or if you have any questions about our licensing terms, please contact us at guidedoa@nature.com.

As you prepare your final submission files, we encourage you to consider whether you have any images or illustrations that may be appropriate for use on the cover of [Nature Genetics / Nature Methods / Nature Physics]. Covers should be both aesthetically appealing and scientifically relevant, and should be supplied at the best quality available. Due to the prominence of these images, we do not generally select images featuring faces, children, text, graphs, schematic drawings, or collages on our covers. We accept TIFF, JPEG, PNG or PSD file formats (a layered PSD file would be ideal), and the image should be at least 300ppi resolution (preferably 600-1200 ppi) in CMYK colour mode. If your image is selected, we may also use it on the journal website as a banner image, and may need to make artistic alterations to fit our journal style. Please submit your suggestions, clearly labelled, along with your final manuscript files. Should we require further information, we will contact you.

Resubmission

Please use the following link for uploading all final materials:

[REDACTED]

*This url links to your confidential homepage and associated information about manuscripts you may have submitted or be reviewing for us. If you wish to forward this e-mail to co-authors, please delete this link to your homepage first.

If you have any questions about the editorial and formatting guidelines, please feel free to contact me. We look forward to receiving your submission within the next 7 days.

Yours sincerely,

Ananya Rastogi, PhD
Associate Editor
Nature Computational Science
orcid.org/0000-0003-3030-8535

Reviewer #1:

Remarks to the Author: Overall significance:

The authors present a mixture of Gaussian process (MoGP) method followed by Dirichlet modeling to predict ALS decline over time in sub-populations or "clusters" across 4 different cohorts. These clusters identify some interesting non-linearities that could be useful for future patient phenotypic once clinical features can be tied to the clusters of progression shapes. The revisions have added much more clarity and better represent the true significance of the findings.

Remarks to the Author: Impact:

The greatest impact of this paper will be in helping the ALS field in trending away from the popular ALSFRS-R linear slope models that, based on the work presented here, have clear limitations due to the quantified non-linearities seen in patient populations.

Remarks to the Author: Strength of the claims:

The strength of the claims have been appropriately adjusted in revision and are now accurately presented in the manuscript.

Remarks to the Author: Reproducibility:

The authors have addressed previous reproducibility concerns with additional experiments, a more clear data availability statement, and supplementary information on the code and methods to accompany the publicly available codebase.

Reviewer #2:

Remarks to the Author: Overall significance:

I am pleased with the current version

Remarks to the Author: Impact:

I am pleased with the current version

Remarks to the Author: Strength of the claims:

I am pleased with the current version

Remarks to the Author: Reproducibility:

I am pleased with the current version

Reviewer #3:

Remarks to the Author: Overall significance:

Copy of my previous comment:

This study provides a characterisation of the longitudinal trajectory of the ALSFRS-R in amyotrophic lateral sclerosis. The developed model was also validated in other datasets. The result is original and can be applied to other fields where longitudinal data is available and behaves non-linear.

Remarks to the Author: Impact:

The authors now added two tutorials which will hopefully improve the ease of application of the developed model.

Remarks to the Author: Strength of the claims:

The authors have their manuscript improved further. It is a pleasure to read it. I will follow the numbers of my previously given comments and the author's replies.

1. Issue solved.

2. Issue solved.

3. Issue solved.

4. Issue solved.

5. Thanks for the additional information. Some issues, however, remain. A). "Slope is calculated using linear regression for all measured data per individual, in points per month", at other places is however another definition used such as decrease in the first year. It would be great if this is consistent. In clinical studies, we usually use (48-score at diagnosis)/time between onset and diagnosis in months. B). The additional information about bulbar onset and age at onset demonstrates potential selection bias. C). Many important variables such as diagnostic delay (or disease duration since onset), forced vital capacity, information about frontotemporal dementia and the C9orf72 repeat expansion (which is present in around 10% of all ALS patients) makes it hard to get a very good feeling for the data used. I understand that this information might be (partially) missing but it would be good to acknowledge this.

6. Issue solved. However, it would be good to mention that the only difference between the ALSFRS and ALSFRS-R is that the ALSFRS-R has two additional questions (and thus 10 identical questions). It would therefore be possible to adapt the model to the ALSFRS but since the ALSFRS-R questionnaire is currently nearly always used (instead of the ALSFRS) I agree with the choice of the authors.

7. Thanks for providing the additional analyses. This highlights the strengths and weaknesses of the different models.

8. Issue solved.
9. Issue solved.
10. Issue solved.
11. The addition of these analyses, figure and table (supp figure 7 & supp table 5) are of great value. Thanks for adding this.
12. Issue solved.
13. Issue solved.
14. I agree with the suggestion of the authors.
15. This supplementary figure 5 is very illustrative. Thanks for adding this figure. Because a lot of space is occupied by the (relatively) rare outliers, the readability could be possibly further increased by a non-linear transformation (such as a square root or log transformation) of the y-axis. Another option could be to add a table with the values of the boxplot (i.e. lower whisker, lower rectangle, medium, upper rectangle, upper whisker).
16. Issue solved.
17. Issue solved.
- 18.

Remarks to the Author: Reproducibility:

Copy of my previous comment:

The analyses were appropriate. The developed model was validated in other datasets. The code for this study is provided online.

Author rebuttal, second version:

Reviewer #1:

Remarks to the Author: Overall significance:

The authors present a mixture of Gaussian process (MoGP) method followed by Dirichlet modeling to predict ALS decline over time in sub-populations or "clusters" across 4 different cohorts. These clusters identify some interesting non-linearities that could be useful for future patient phenotypic once clinical features can be tied to the clusters of progression shapes. The revisions have added much more clarity and better represent the true significance of the findings.

Remarks to the Author: Impact:

The greatest impact of this paper will be in helping the ALS field in trending away from the popular ALSFRS-R linear slope models that, based on the work presented here, have clear limitations due to the quantified non-linearities seen in patient populations.

Remarks to the Author: Strength of the claims:

The strength of the claims have been appropriately adjusted in revision and are now accurately presented in the manuscript.

Remarks to the Author: Reproducibility:

The authors have addressed previous reproducibility concerns with additional experiments, a more clear data availability statement, and supplementary information on the code and methods to accompany the publicly available codebase.

Reviewer #2:

Remarks to the Author: Overall significance:

I am pleased with the current version

Remarks to the Author: Impact:

I am pleased with the current version

Remarks to the Author: Strength of the claims:

I am pleased with the current version

Remarks to the Author: Reproducibility:

I am pleased with the current version

Reviewer #3:

Remarks to the Author: Overall significance:

Copy of my previous comment:

This study provides a characterisation of the longitudinal trajectory of the ALSFRS-R in amyotrophic lateral sclerosis. The developed model was also validated in other datasets. The result is original and can be applied to other fields where longitudinal data is available and behaves non-linear.

Remarks to the Author: Impact:

The authors now added two tutorials which will hopefully improve the ease of application of the developed model.

Remarks to the Author: Strength of the claims:

The authors have their manuscript improved further. It is a pleasure to read it. I will follow the numbers of my previously given comments and the author's replies.

1. Issue solved.

2. Issue solved.

3. Issue solved.

4. Issue solved.

5. Thanks for the additional information. Some issues, however, remain. A). "Slope is calculated using linear regression for all measured data per individual, in points per month", at other places is however another definition used such as decrease in the first year. It would be great if this is consistent. In clinical studies, we usually use (48-score at diagnosis)/time between onset and diagnosis in months.

We thank the reviewer for raising this issue, which we have now clarified.

We now use two version of slopes, and describe the purpose of each. In the revised version, anytime we calculate a slope from raw datapoints (Fig 2-4, Supp. Table 1), we use the same definition of slope. The slope is the anchored linear regression for all available data points per individual, where a score of 48 at symptom onset has been imputed. This measure is similar to the reviewer's recommended measure, but here, all data points are used instead of just the score at diagnosis. One benefit of calculating the slope this way over the definition the reviewer provides (which uses only the score at diagnosis) is that it is less susceptible to outliers, because it calculates a function using more data-points per individual.

In Figure 1, we calculate an additional anchored first-year slope for each cluster. This is calculated as follows: (48- cluster mean function at one year)/time from symptom onset. Here, the analysis intends to evaluate how generalizable the cluster trajectories in the first year of disease progression are to subsequent timepoints. We think this presents an interesting result with clinical applications to studies which use the first year slope (such as the PRO-ACT Dream Challenges). We have changed the term to be clear that these are different analysis; we now specify this as a "first year slope", so as to not be confused with other measures used in the text.

B). The additional information about bulbar onset and age at onset demonstrates potential selection bias.

We have added the following to the supplementary discussion:

The variation in ages of onset and prevalence of sites of onset differ across clinical cohorts, which can indicate additional potential selection biases.

C). Many important variables such as diagnostic delay (or disease duration since onset), forced vital capacity, information about frontotemporal dementia and the C9orf72 repeat expansion (which is present in around 10% of all ALS patients) makes it hard to get a very good feeling for the data used. I understand that this information might be (partially) missing but it would be good to acknowledge this.

We have added the following to the supplementary discussion:

Other variables that can be used to evaluate selection bias but were partially missing or unavailable across our studies include diagnostic delay, forced vital capacity, frontotemporal dementia, and c9orf72 status.

6. Issue solved. However, it would be good to mention that the only difference between the ALSFRS and ALSFRS-R is that the ALSFRS-R has two additional questions (and thus 10 identical questions). It would therefore be possible to adapt the model to the ALSFRS but since the ALSFRS-R questionnaire is currently nearly always used (instead of the ALSFRS) I agree with the choice of the authors.

We have added the following to the methods:

The ALSFRS-R is an updated version of the previously used ALSFRS metric, which adds additional questions measuring dyspnea, orthopnea, and respiratory insufficiency.⁶ The ALSFRS-R measure was used here because it is the current standard in clinical trial analysis.¹²⁻¹⁴

7. Thanks for providing the additional analyses. This highlights the strengths and weaknesses of the different models.

8. Issue solved.

9. Issue solved.

10. Issue solved.

11. The addition of these analyses, figure and table (supp figure 7 & supp table 5) are of great value. Thanks for adding this.

12. Issue solved.
13. Issue solved.
14. I agree with the suggestion of the authors.
15. This supplementary figure 5 is very illustrative. Thanks for adding this figure. Because a lot of space is occupied by the (relatively) rare outliers, the readability could be possibly further increased by a non-linear transformation (such as a square root or log transformation) of the y-axis. Another option could be to add a table with the values of the boxplot (i.e. lower whisker, lower rectangle, medium, upper rectangle, upper whisker).
We have applied a square root transformation to this plot. We have also replaced the main text figure 4 with supp. figure 5, to comply with Nature Computational Science data presentation requirements.

16. Issue solved.
17. Issue solved.

Remarks to the Author: Reproducibility:

Copy of my previous comment:

The analyses were appropriate. The developed model was validated in other datasets. The code for this study is provided online.

Final Decision Letter:

Subject: Acceptance of GUIDEDOA-21-00225B
Message: 14th July 2022

Dear Professor Fraenkel,

We are pleased to inform you that your manuscript entitled "Identifying Patterns of ALS Progression from Sparse Longitudinal Data" is now accepted for publication in Nature Computational Science. Thank you again for choosing to be a part of Guided Open Access at the Nature Portfolio.

Prior to typesetting your manuscript, we may make minor changes to enhance the lucidity of the text and with reference to our house style. In approximately two weeks you will receive a link to our online eProof. We ask that you please read your proof with care to ensure that we have not inadvertently introduced any errors, or altered the sense of your paper in any way. Please note that the corresponding author is responsible on behalf of all co-authors for the accuracy of all content, including spelling of names and current affiliations. If you have any upcoming travel or conflicts that may make you unavailable to perform these checks in the near future please contact us immediately.

You will be notified once your paper is scheduled for publication and our Press team may distribute a press release to news organizations worldwide, which may include details of your work. We are happy for your institution or funding agency to prepare its own press release, but we encourage your Public Relations or Press Office to contact ours at press@nature.com. Please include your manuscript tracking number (GUIDEDOA-21-00225B) and the name of the journal when you contact us.

As part of Springer Nature's commitment to transparency into our peer review process, all papers published out of the Guided Access initiative will formally acknowledge our reviewers for the time and attention that they provide our editorial process. All peer-reviewed content will carry an anonymous statement of peer reviewer acknowledgement, and for those reviewers who give their consent, we will publish their names alongside the published article. For more information, please refer to our FAQ page:

<https://www.nature.com/documents/guidedoa-reviewer-information.pdf>. Additionally, when your paper is published, we will include a note in the Peer Review section highlighting that it was reviewed as part of the Guided Open Access initiative.

You can now use a single sign-on for all your accounts, view the status of all your manuscript submissions and reviews, access usage statistics for your published articles and download a record of your refereeing activity for the Nature Portfolio journals.

We thank you again for being a part of Guided Open Access, and we look forward to publishing your paper.

Best regards,

Ananya Rastogi, PhD
Associate Editor
Nature Computational Science
orcid.org/0000-0003-3030-8535

** See Nature Portfolio's author and referees' website at www.nature.com/authors for information about policies, services and author benefits